# 4-Acetyl-Antroquinonol B Improves the Sensitization of Cetuximab on Both Kras Mutant and Wild Type Colorectal Cancer by Modulating the Expression of Ras/Raf/miR-193a-3p Signaling Axis

**DOI:** 10.3390/ijms22147508

**Published:** 2021-07-14

**Authors:** Yi Cheng Chu, Tung-Yao Tsai, Vijesh Kumar Yadav, Li Deng, Chun-Chih Huang, Yew-Min Tzeng, Chi-Tai Yeh, Ming-Yao Chen

**Affiliations:** 1Department of Medicine, School of Medicine, St. George’s University, St. George SW17 0RE, Grenada; ychu@sgu.edu; 2Department of Emergency Medicine, Taipei Medical University-Shuang Ho Hospital, New Taipei City 23561, Taiwan; 11325@s.tmu.edu.tw; 3Graduate Institute of Injury Prevention and Control, Taipei Medical University, Taipei 110, Taiwan; 4Department of Emergency Medicine, School of Medicine, Taipei Medical University, Taipei 110, Taiwan; 5Division of Gastroenterology and Hepatology, Department of Internal Medicine, School of Medicine, College of Medicine, Taipei Medical University, Taipei 110, Taiwan; 20604@s.tmu.edu.tw; 6Division of Gastroenterology and Hepatology, Department of Internal Medicine, Taipei Medical University-Shuang Ho Hospital, New Taipei City 23561, Taiwan; 7Beijing Bioprocess Key Laboratory, College of Life Science and Technology, Beijing University of Chemical Technology, Beijing 100029, China; dengli@mail.buct.edu.cn; 8Amoy-BUCT Industrial Bio-Technovation Institute, Xiamen 361022, China; 9Center for General Education, National Taitung University, Taitung 950, Taiwan; john@newbellus.com.tw (C.-C.H.); tzengym@gmail.com (Y.-M.T.); 10Department of Medical Research & Education, Taipei Medical University-Shuang Ho Hospital, New Taipei City 23561, Taiwan; 11Department of Medical Laboratory Science and Biotechnology, Yuanpei University of Medical Technology, Hsinchu 300, Taiwan

**Keywords:** 4-AAQB, cetuximab, KRAS mutation, resistance, colorectal cancer

## Abstract

The KRAS mutation is one of the leading driver mutations in colorectal cancer (CRC), and it is usually associated with poor prognosis and drug resistance. Therapies targeting the epidermal growth factor receptor (EFGR) are widely used for end-stage CRC. However, patients with KRAS mutant genes cannot benefit from this therapy because of Ras signaling activation by KRAS mutant genes. Our previous study revealed the anti-proliferative effect of 4-acetyl-antroquinonol B (4-AAQB) on CRC cells, but whether the drug is effective in KRAS-mutant CRC remains unknown. We screened CRC cell lines harboring the KRAS mutation, namely G12A, G12C, G12V and G13D, with one wild type cell line as the control; SW1463 and Caco-2 cell lines were used for further experiments. Sulforhodamine B assays, together with the clonogenicity and invasion assay, revealed that KRAS-mutant SW1463 cells were resistant to cetuximab; however, 4-AAQB treatment effectively resensitized CRC cells to cetuximab through the reduction of colony formation, invasion, and tumorsphere generation and of oncogenic KRAS signaling cascade of CRC cells. Thus, inducing cells with 4-AAQB before cetuximab therapy could resensitize KRAS-mutant, but not wild-type, cells to cetuximab. Therefore, we hypothesized that 4-AAQB can inhibit KRAS. In silico analysis of the publicly available GEO (GSE66548) dataset of KRAS-mutated versus KRAS wild-type CRC patients confirmed that miR-193a-3p was significantly downregulated in the former compared with the latter patient population. Overexpression of miR-193a-3p considerably reduced the oncogenicity of both CRC cells. Furthermore, KRAS is a key target of miR-193a-3p. In vivo treatment with the combination of 4-AAQB and cetuximab significantly reduced the tumor burden of a xenograft mice model through the reduction of the expression of oncogenic markers (EGFR) and p-MEK, p-ERK, and c-RAF/p-c-RAF signaling, with the simultaneous induction of miR-193a-3p expression in the plasma. In summary, our findings provide strong evidence regarding the therapeutic effect of 4-AAQB on KRAS-mutant CRC cells. Furthermore, 4-AAQB effectively inhibits Ras singling in CRC cells, through which KRAS-mutant CRC can be resensitized to cetuximab.

## 1. Introduction

Cancer is one of the major causes of death worldwide [1]. Among organ cancers, gastrointestinal (GI) tract cancer is the major cause of mortality in patients, particularly colorectal cancer (CRC) [2]. CRC is the third ubiquitous cancer worldwide and in Asia [3]. For a long time, sporadic CRC has been perceived as a homogenous condition that occurs with the adenoma-carcinoma phenotype, which is caused by many genetic alterations of CRC key genes, such as KRAS, tumor suppressor protein (TP) 53, and APC Regulator Of WNT Signaling Pathway (APC) [4]. However, with breakthrough genetic technologies providing a clear understanding of the driver mutation effect in different cancer types, we are entering the era of precision medicine [5]. KRAS-mutant CRC is associated with poor prognosis (high rate of metastasis and incidence of therapy resistance) [6].

As the KRAS mutation is the major driver mutation in CRC [7], developing a drug targeting the Ras pathway is urgently needed. Recently, a promising new drug called Sotorasib (AMG 510), which is a novel KRAS inhibitor targeting the KRAS G12C mutation, was approved for treating patients with metastatic non-small-cell lung cancer harboring the KRAS G12C mutation [8]. This illustrates that developing Ras inhibitors for treating cancer is important, and it might become the next step in cancer treatment.

The mutation status of the KRAS gene has been observed to affect the response of CRC toward cetuximab treatment, cetuximab a monoclonal antibody that binds to the extracellular region of EGFR, is effective on KRAS wild type metastatic CRC [9,10,11]. This drug demonstrates anticancer effects through the suppression of EGFRs; it was developed in the 1970s and plays a crucial role in metastatic colorectal cancer [12]. However, patients with KRAS-mutant CRC do not benefit from this treatment [9]. Several studies have illustrated that Ras signaling overexpression can compensate for EGFR inhibition, causing the failure of anti-EGFR treatment in KRAS-mutant CRC [13]. Thus, the blockage of the Ras signal pathway can resensitize KRAS-mutant CRC cell lines to anti-EGFR effect treatment.

*Antrodia cinnamomea* is a unique fungus, which is exclusively found in Taiwan. It is traditionally known to have anticancer properties. In a broad spectrum of cancers, 4-acetyl-antroquinonol B (4-AAQB) isolated and purified from *A. cinnamomea* exerts anti-proliferative effects [14,15,16]. Our previous study illustrated the anti-CRC role of 4-AAQB, which is mediated through the inhibition of the formation of CRC cancer stem cells and reactive oxygen species (ROS) oxidative stress, resulting in the modulation of CRC cells’ innate or acquired insensitivity towards chemotherapy [17,18]. Growing evidence exists of the role of small non-coding RNA, particularly micro-RNA (miRNA’s), in controlling the key biological process, including deciding the fate of cancer treatment [19], such as the tumor-suppressive effect of miR-193a-3p in lung cancer through targeting KRAS expression [20,21]. As the sequel of our previous published work, in this study, we specifically targeted CRC cells harboring the KRAS mutation and the results showed that KRAS-mutant CRC cells are adequately resistant to cetuximab (anti-EGFR monoclonal antibodies). In vitro studies of 4-AAQB alone, or in combination have demonstrated significant anti-cancer effects on KRAS mutant CRC cell lines. Our study results, both in silico and in vitro, suggested that the miR-193a-3p expression could predict the potential response of KRAS-mutant CRC cells to the treatment. Targeting KRAS-mutant CRC cells and mice xenograft model with 4-AAQB results in the over-expression of miR-193a-3p and the reduction of CRC tumorigenesis. In summary, both in vivo and in vitro studies indicate that 4-AAQB may be an important therapeutic agent that targets KRAS-mutant CRC cells through the reduction of the Ras-signaling cascade and modulation of the expression of key miRs in CRC tumorigenesis.

## 2. Results

### 2.1. Screening and Patient Survival Analysis in Patients with KRAS Mutation and Wild-Type Genes

In the CRC patient cohort from the TCGA database, the missense KRAS mutation was observed significantly higher in late-stage patients as compared to the patients with wild-type KRAS (Figure 1A). KRAS-mutant CRC is associated with a poorer prognosis compared to KRAS wild-type CRC, and the KRAS G12C mutation has the worst prognosis among KRAS mutations. In the metastatic CRC (MSKCC, Cancer Cell 2018) dataset, we noticed that patients harboring KRAS mutations had a poor prognosis based on analysis with cBioPortal (Figure 1B). Overall survival at 10-year and 15-year time-points postdiagnosis in patients with KRAS mutation (*n* = 507) were 32% and 28% and in KRAS wild-type (*n* = 627) patients were 42% and 36%, respectively (*p* < 0.05). KRAS mutation subgroups, namely G12A (*n* = 26), G12C (*n* = 32), G12V (*n* = 100), and G13D (*n* = 90) were analyzed (Figure 1C). Although the G12C KRAS mutation has the poorest prognosis (10-year overall survival: 18%), no statistical significance was reached. The CRC cell lines screened were Caco-2, SW1116, SW1463, SW620, HCT116, and DLD-1. Four cancer cell lines originate from the colon, namely Caco-2, SW1116, SW620, and DLD-1. SW1463 and HCT116 are isolated from the rectum and ascending colon, respectively. Both HCT116 and DLD-1 harbor KRAS G13D mutation, whereas, SW1116, SW1463, and SW620 harbor KRAS G12A, KRAS G12C, and KRAS G12V mutations, respectively. We used Caco-2 cells (KRAS wild type) and SW1463 cells (KRAS-mutation; G12C) for our study (Figure 1D).

### 2.2. KRAS-Mutated Colorectal Cells Were Resistant to Cetuximab Treatment 

KRAS-mutated CRC cells are frequently resistant to anti-EGFR treatments, the protein expression level of the p-EGFR/EGFR in KRAS-wild (Caco-2) and KRAS-mutant CRC cell lines (Figure 2A). The dose and time-dependent effects of cetuximab on cell proliferation were assessed through the SRB assay. As shown in Figure 2B,C, KRAS-mutated SW1463 (G12C) cells showed higher resistance to cetuximab in comparison to other KRAS-mutant CRC cells both dose/time-dependent, whereas maximum inhibition in cell proliferation of the KRAS-wild type Caco-2 cells was achieved at the given concentration of cetuximab. Complete inhibition of SW1463 cell proliferation could not be reached with 1000 µg/mL of cetuximab. Thus, anti-EGFR antibody treatment was effective for KRAS-wild Caco-2 cells, but single-drug therapy of cetuximab was not sufficient to achieve a therapeutic outcome on KRAS-mutant CRC cells. Cetuximab treatment did not affect the morphology of the SW1463 cells (cell shrinkage, cytoplasmic membrane blebbing, and cell death) compared with the control untreated cells (Figure 2D). However, Caco-2 cells were sensitive to cetuximab. We next performed a clonogenic and invasion assay with the indicated treatments. In this assay, cetuximab treatment for Caco-2 cells displayed a 50–60% inhibition in colony-forming and invasive properties (Figure 2E,F). By contrast, no change in the clonogenic and invasive property was noticed for SW1463 cells (Figure 2E,F).

### 2.3. 4-AAQB Treatment Results in Inhibition of Oncogenic Properties of KRAS-Mutant and Wild-Type Colorectal Cells

We determined whether 4-AAQB (Figure 3A) treatment could overcome the therapy-resistance of KRAS-mutant CRC cells. A cell proliferation study of CRC cell lines based on the effect of 4-AAQB treatment was conducted using the SRB assay. A dose-dependent effect was observed at both time points that is, 24 and 48 h of treatment (Figure 3B). After 4-AAQB treatment for 48 h, 50% inhibition of cell proliferation occurred for all CRC cells within approximately IC_50_ of 15 μM (Figure 3C). Furthermore, SW1463 and Caco-2 cell morphology examination under the microscope showed that 4-AAQB treatment effectively induced apoptosis in both cells (Figure 3D). Next, the effect of 4-AAQB on the invasion and colony-forming abilities of CRC cells (SW1463 and Caco-2) was investigated. Under IC_25_ treatment with 4-AAQB for 48 h, the invasive (Figure 3E) and colony-forming (Figure 3F) abilities of the cells were greatly inhibited, indicating that 4-AAQB effectively reduced the mobility and invasiveness of the CRC cells, compared with their untreated control counterparts. Furthermore, for determining the effect of 4-AAQB on CRC cell tumorigenesis, we assayed the colon-sphere formation of CRC cells (SW1463 and Caco-2). Tumor-sphere formation assay is crucial for the identification of stemness and drug resistance [22,23]. The tumorsphere formation abilities of SW1463 and Caco-2 were effectively suppressed by 4-AAQB (Figure 3G). Treatment of SW1463 and Caco-2 cells with 4-AAQB resulted in the reduction of RAF/MEK/ERK signaling and Ras cascade, as shown by a decrease in the protein levels of p-MEK, p-ERK, c-RAF, p-C-RAF, and RAS (Figure 3H). Furthermore, as shown in Appendix A, reduction in the oncogenic marker expression (p-mTOR, p-AKT, p-PI3K), and B-RAF/p-B-RAF was observed after the 4-AAQB treatment on both the CRC cells (SW1463 and Caco-2). Thus, 4-AAQB plays a pivotal role in the sensitization of KRAS-mutant SW1463 cells to the therapy.

### 2.4. Combination Treatment with 4-AAQB and Cetuximab Increased Cetuximab Sensitivity in KRAS-Mutant CRC Cells

Epidermal growth factor receptor (EGFR), a protein tyrosine kinase receptor, is frequently expressed in CRC and is involved in cell proliferation and cell survival [24]. Anti-EGFR therapy including cetuximab and panitumumab significantly improves the survival of KRAS wild-type MSKCC, but not of those with KRAS-mutant cancer [25], suggesting that the single-drug therapy of cetuximab is insufficient to achieve a therapeutic effect. To qualitatively analyze whether the combination of 4-AAQB and cetuximab could produce synergistic anti-proliferative effects, a combination index (CI) is calculated. CI values at IC_50_ points were calculated according to the results of SRB assays and the CI values are shown in Figure 4A. Treatment with 4-AAQB in combination with cetuximab exhibited an overall synergistic effect (Figure 4B). In KRAS wild-type Caco-2 cells, low-dose (3–15 μM) 4-AAQB demonstrated a slight synergistic effect, but when the 4-AAQB concentration was 12 μM in combination with 5 μM cetuximab, the CI value increased to 1, which indicates only an additive effect at this concentration. Moreover, increasing the concentration of 4-AAQB to 15 μM decreased the CI value to 0.72–0.75, indicating a moderate synergistic effect of this drug in combination with cetuximab. In KRAS mutant-SW1463 cells, the synergistic effect leads to a re-boost. With low-dose (3 μM) 4-AAQB, the CI value ranges from 0.6 to 0.72, demonstrating moderate synergism of the drug in combination with cetuximab. Furthermore, increasing the concentration of 4-AAQB to 3–15 μM decreased the CI value from 0.24 to 0.34, which indicates strong synergism. In the high-dose 4-AAQB group, the CI value further decreased to <0.2 (0.16–0.19), indicating an excellent synergistic effect. Thus, we also tested whether the combination of 4-AAQB and cetuximab can remarkably suppress CRC colony formation, tumorsphere generation, and cell proliferation abilities. Strikingly, the combination of 4-AAQB and cetuximab treatment synergistically resensitized cells and inhibited CRC colony formation, tumorsphere generation, and cell proliferation abilities through the induction of apoptosis (Figure 4C–E). To further understand whether the anti-proliferative effects on cell proliferation involve apoptotic machinery, SW1463 and Caco-2 cells were treated with 9–12 µM 4-AAQB or 2.5–5 µM cetuximab or their combination for 24 h; then, protein assays were conducted to determine the expression level of cleaved (cl) caspase-3/9 and PARP. Treatment with 4-AAQB and cetuximab alone enhanced the expression of cl-caspase-3 and cl-PARP as compared with the control (Figure 4F). Combination treatment induced a more than twofold increase in the expression of cl-caspase-3 and cl-PARP in SW1463. These results illustrated that the synergistic anti-proliferative effect of 4-AAQB and cetuximab is through cl-caspase-3/9 and cl-PARP expression on CRC cells. Together with the induction of expression apoptotic markers, combination treatment effectively modulates the expression of activated p-EGFR-B-raf/c-Raf-Erk-Mek, as described in the Western blot image (Figure 4F).

### 2.5. Anti-CRC Role of 4-AAQB Was Associated with the Induction of miR-193a-3p and Reductions of Its Oncogenic Targets

Noncoding RNAs, particularly miRs, have gained much attraction in the field of oncology. We analyzed a publicly available database from GEO (GSE66548) (Figure 5A). Heatmaps revealed that miR-193a-3p, one of the 25 miR, expression was significantly lower in the KRAS-mutant CRC patient samples (*n* = 15) than in the KRAS wild-type counterpart (*n* = 15). Furthermore, after treating CRC cells (SW1463 and Caco-2) with 4-AAQB, quantitative reverse transcription-polymerase chain reaction (qRT-PCR) analysis showed that the expression of miR-193a-3p was significantly higher in the treatment group than in the control group (Figure 5B). We evaluated the effect of miR-193a-3p in KRAS-mutant cancer cells by using gain-and-loss-of-function experiments (Figure 5C). Compared with the control group, miR-193a-3p overexpression significantly inhibited the migration (Figure 5D), colony formation (Figure 5E), and cell proliferation (Figure 5F) abilities of Caco-2 and SW1463 cells after transfection with miR-193a-3p (mimic). We then searched for miR-193a-3p targets and identified KRAS as a key target, as evidenced by the binding of 3′-untranslated regions to miR-193a-3p (Figure 5G). Consistently, KRAS was the molecular target of miR-193a-3p, as demonstrated through Western blots of SW1463 and Caco-2 cells transfected with miR-193a-3p mimic and inhibitor molecules. Exogenous miR-193a-3p mimic molecules led to decreased expressions of the predicted targets, namely KRAS, and inhibition of the associated downstream molecules of p-MEK, p-ERK, c-RAF, and PARP1 in both SW1463 and Caco-2 cells (Figure 5H); contrary observations were made when cells were transfected with the miR-193a-3p inhibitor. Notably, treatment with 4-AAQB resulted in the increased expression of miR-193a-3p and target KRAS through this axis.

### 2.6. Treatment with 4-AAQB Increased Cetuximab Efficacy In Vivo

After establishing the anti-CRC role of 4-AAQB in vitro, for our in vivo study, we evaluated the effect of 4-AAQB by using a xenograft mouse SW1463 tumor model. The tumor size and volume over time clearly showed that combination treatment with 4-AAQB and cetuximab significantly delayed tumorigenesis followed by 4-AAQB and cetuximab treatment alone compared with vehicle (Figure 6A,B). Western blot analysis of tumor samples collected from all groups demonstrated the phosphorylated, and of non-phosphorylated protein levels expression of oncogenic markers (p-EGFR), p-ERK, p-MEK, c-RAF/BRAF and p-c-RAF/BRAF (Figure 6C), whereas the qRT-PCR analysis of blood-plasma levels showed the highest level of miR-193a-3p expression in combination-treated pooled blood samples, followed by 4-AAQB, cetuximab, and vehicle control (Figure 6D). As expected, IHC analysis of combination treatment indicated that the expression of activating c-RAF protein level was effectively reduced, whereas, the expression of apoptotic markers (p-c-RAF, cl-Caspase3 and cl-PARP) was induced in the combination treatment group as compared to 4-AAQB alone and cetuximab treatment and vehicle control (Figure 6E) resulting in the reduction of tumor burden in the xenograft mouse model.

## 3. Discussion

Cancers of the GI tract, mainly CRC, are a major cause of mortality in patients [2] and the third commonest diagnosed cancer worldwide and in Asia [3]. Intrinsic and acquired therapy resistance are the main hurdles to overcome for maximizing the benefits of therapy. Notably, the KRAS mutation is associated with low overall survival in patients with advanced-stage CRC who are treated with cetuximab after radiation therapy [27]. The cetuximab and panitumumab anti-EGFR therapy had significantly improved the survival of KRAS wild-type MSKCC patients, but for the KRAS mutant group, it is ineffective [25]. Previous studies have demonstrated the anti-EGFR effect is Ras-dependent [28]. Overexpression of the Ras-signaling pathway in KRAS-mutant CRC disrupts the downstream signal transduction of anti-EGFR therapy that renders KRAS-mutant CRC resistant to cetuximab [29]. Therefore, KRAS-mutant CRC can serve as a good research model to study the molecular mechanism of anti-EGFR therapy as well as to establish and discover new therapeutic strategies for reverting the resistance of KRAS-mutant CRC to cetuximab.

In this study, we examined the effect of KRAS-mutation status on the overall survival among CRC patients (Figure 1) and showed the resistance of KRAS-mutated CRC cells to cetuximab (Figure 2A). The cell proliferation assay showed that KRAS-mutant SW1463 cells were cetuximab-resistant in comparison with Caco-2 cells (Figure 2B). Furthermore, no change in the clonogenic and invasive properties of SW1463 cells was noted after cetuximab treatment, indicating their resistance to cetuximab (Figure 2D,F).

Drug resistance caused by the overexpression of the Ras-signaling pathway frequently occurs in KRAS-mutant CRC [30]. Inhibiting the Ras-signaling is crucial to overcome its drug resistance [31]. Thus, we hypothesized that modulation of the Ras signaling pathway by using 4-AAQB could resensitize KRAS-mutant CRC cells to anti-EGFR therapy. Our present work is the first to suggest that 4-AAQB treatment effectively targets KRAS-mutant and wild-type CRC cells and resensitizes them to cetuximab therapy. The 4-AAQB treatment effectively reduced the proliferation of CRC cells dose-dependently along with the reduction in the oncogenic property and tumor-sphere generation abilities of CRC cells (SW1463 and Caco-2) (Figure 3). Protein expression revealed that 4-AAQB treatment reduced the phosphorylation activation of key targets participating in the Raf/Mek/Erk pathways and RAS cascade (Figure 3H) [32]. This is consistent with our previous study, which showed that 4-AAQB suppressed tumorigenesis and inhibited cancer stem cell-like phenotype through multiple signaling pathways, including JAK-STAT and Wnt/β-catenin [17]. Moreover, literature demonstrated that 4-AAQB induced the apoptosis of CRC cells through the suppression of the Ras-signaling pathway via modulating the expression of cl-Caspase3/9 and cl-PARP [33]. Furthermore, few studies have reported that increased autophagy is associated with anti-EGFR drug resistance. Previous studies discussed the crosstalk between EGFR signaling and autophagy. Adding autophagy inhibitors may help treat these tumors, and the evaluation of the success of autophagy inhibitory strategies will depend on the improvement of the patient’s prognosis after treatment [34,35]. Our previous study reported the effect of 4-AAQB on ovarian cancer [36]; 4-AAQB could disrupt autophagy through the inhibition of the upstream signal of ATG (autophagy-related gene).

Importantly, for the first time to our knowledge it was demonstrated that through the inhibition of the Ras-signaling pathway, 4-AAQB can inhibit cancer cell growth and proliferation and induces apoptosis. Ras-signaling is involved in anti-EGFR therapy because Ras-protein overexpression causes the inhibitory signal from EGFR [37]. For validation, we comparatively assessed the effect of 4-AAQB and cetuximab as single-drug and in combination in KRAS wild-type Caco-2 cells human CRC cell and KRAS-mutant SW1463 human CRC cells (Figure 4). Our result demonstrated that the combination treatment with 4-AAQB and cetuximab could synergistically overcome the resistance of KRAS-mutant CRC cells (SW1463) and KRAS-wild (Caco-2) cells to cetuximab. These findings illustrate that 4-AAQB inhibits Ras-signaling through phosphorylation modification of the RAF/MEK/ERK and Ras cascade, enhancing the potency of anti-EGFR therapy. Furthermore, regarding epigenetic factors controlling the severity of KRAS-mutant CRC cells, we found that miR-193a-3p expression was effectively downregulated in patients with KRAS-mutant CRC (Figure 5). Furthermore, a previous study also suggested that miR-193-3p inhibition in CRC patients regulated tumor progression [38]. Conversely, miR-193a-3p overexpression inhibited tumor progression. Both in vitro and in vivo, 4-AAQB alone or in combination with cetuximab increased miR-193a-3p expression in the xenograft mice model (Figure 5 and Figure 6).

As EGFR is the upstream initiator of the Ras pathway, inhibiting EGFR cannot effectively inhibit downstream signals when Ras drives the whole pathway [39]. Resensitization of KRAS-mutant CRC to cetuximab after turning off Ras signaling has been reported in several studies [40]. 4-AAQB, which shares its structure to antroquinonol (AQ), has demonstrated good anticancer potency in CRC [41]. Both 4-AAQB and AQ suppress Ras-signaling, but which drug has a stronger effect on CRC remains elusive. We hope that the use of the natural compounds of 4-AAQB would benefit many patients with CRC harboring KRAS mutations.

## 4. Materials and Methods

### 4.1. Clinical Sample Collection and Preparation

Written informed consent for the collection of clinical colon cancer samples was obtained from all participants, and this study was approved by the Medical Ethics Committee of Taipei Medical University-Joint Institutional Review Board (N202104054; Taipei, Taiwan). The samples were then analyzed and confirmed by two independent pathologists.

### 4.2. Drugs, Chemicals, Cell Culture and Media

Erbitux (the active ingredient of cetuximab) was provided by Taipei Medical University-Shuang Ho Hospital (Taiwan) and was originally purchased from Merck KGaA (Darmstadt, Germany), a stock solution of 5 mg/mL was stored at 4 °C. Furthermore, 4-AAQB (>99% HPLC purity) was purchased from New Bellus Enterprises Co., Ltd. (Tainan, Taiwan), a stock solution of 10 mM was stored at −20 °C. Dulbecco’s modified Eagle’s medium (DMEM; GibcoTM, 12-634-010), Leibovitz’s L-15 (L-15), and Minimal Essential Medium (MEM) were purchased from Thermo Fisher Scientific Inc., Waltham, MA, USA. Furthermore, Gibco^®^ RPMI 1640, fetal bovine serum (FBS), Trypsin/ethylenediaminetetraacetic acid (EDTA), dimethyl sulfoxide (DMSO), phosphate-buffered saline (PBS), sulforhodamine B (SRB) medium, acetic acid, and TRIS base were also purchased from Sigma-Aldrich Co. (St. Louis, MO, USA).

### 4.3. Cell Lines

Human CRC cell lines HCT116 (ATCC-CCL-247, Kras G13D mutation), DLD-1 (ATCC-CCL-221, Kras G13D mutation), SW1463 (ATCC-CCL-234, Kras G12C mutation), SW620 (ATCC-CCL-227, Kras G13V mutation), Caco-2 (ATCC-HTB-37, Wild-Type KRAS), and SW1116 (ATCC-CCL-233, Kras G12A mutation) were purchased from the American Type Culture Collection (ATCC, Manassas, VA, USA). HCT116, DLD-1, and SW620 were cultured in DMEM; SW1116 and SW1463 were cultured in Leibovitz’s L-15 medium and Caco-2 was cultured in MEM medium, respectively. All media were supplemented with 10% FBS and 1% penicillin/streptomycin (Invitrogen, Life Technologies, and Carlsbad, CA, USA). Cells were cultured at 95% confluence and the medium was changed every 72 h and incubated at 37 °C in a 5% humidified CO_2_ incubator. Cells were treated with different concentrations of 4-AAQB and cetuximab for different durations.

### 4.4. In Silico Data Acquisition and Analysis

To further investigate the biological effect of the KRAS mutation in CRC patients, the data from The Cancer Genome Atlas Program (TCGA)-CRC and Metastatic CRC (MSKCC, Cancer Cell 2018) dataset contains survival data with the clinical information, KRAS-mutation, and mRNA-expression values were downloaded and analyzed from cBioPortal for Cancer Genomics (http://www.cbioportal.org/, accessed on 20 July 2020). The genome-wide miRNA expression level of the 30 CRC patients’ data between KRAS-wild type and KRAS-mutant tumors samples were download from the publicly available Gene Expression Omnibus (GEO) database (https://www.ncbi.nlm.nih.gov/geo/, accessed on 21 July 2020), and the accession numbers are GSE66548 [42]. We used the edgeR and pheatmap package from the R software (R Version 3.3.2) to perform the differential analysis (http://www.bioconductor.org/packages/release/bioc/html/edgeR.html, accessed on 15 July 2020) and heatmap cluster analysis (https://cran.r-project.org/web/packages/pheatmap/index.html, accessed on 20 July 2020).

### 4.5. Colorimetric Assay of Sulforhodamine B (SRB) Cell Proliferation Assay

CRC cells were seeded in a supplemented medium at a density of 3000 cells/well in triplicate in 96-well plates and incubated at 37 °C in 5% humidified CO_2_ for 24 h. Monotherapies of 4-AAQB and cetuximab were performed for 48 and 72 h. Treated cells were washed with PBS and trichloroacetic (TCA) was added to fix the viable cells for 1 h at 4 °C. Then, cells were washed with distilled water and viable cells were incubated in 0.4% sulforhodamine B (SRB) for 1 h at room temperature, unbounded SRB dye was purged with 1% acetic acid washing twice. Plates were put in the incubator for air-drying; the attached dye was dissolved in a 10 mM trizma base (Tris). Absorbance was read in a microplate reader at a wavelength of 565λ. Caco-2 and SW1463 were selected for further experiments on 4-AAQB and cetuximab combination therapy. For the combination therapy, the combination index (CI) theorem of Chou-Talalay’s quantitative definition for the evaluation of combination effects [43] was applied on CRC cells. Caco-2 and SW1463 cells were seeded at a density of 3000 cells/well and incubated for 24 h; 4-AAQB was added first for induction (24-h incubation) followed by cetuximab and cells were treated for another 24 h.

### 4.6. Western Blotting

CRC cells were cultured at a density of 5 × 10^6^ and then trypsinized to detach the adherent cells after the respective treatment. Cells were lysed using RIPA buffer (Cell Signaling Technology) with a cocktail of protease inhibitors (Sigma Aldrich). Then, 10 μg of protein samples were subjected to immunoblotting and blocked with 3% BSA in TBST for 1 h at room temperature and incubated and shaken at 4 °C with respective primary antibody overnight. All primary and secondary antibodies used, along with their dilutions and molecular weight, are listed in Appendix A. Membranes were then washed with TBST twice for 15 min and incubated with horseradish peroxidase (HRP)-labeled secondary antibody for 1 h at room temperature. Bands were detected using the Western blotting reagents and the BioSpectrum Imaging System (Ultra-Violet Products Ltd., Upland, CA, USA).

### 4.7. Transient Oligonucleotide Transfection

MiR-193a-3p (Inhibitor and mimic) and negative control miRNA were purchased from ThermoFisher Scientific (USA) and prepared under strict adherence to the vendor’s instructions. Cells were transfected using Lipofectamine^®^ 2000 (Invitrogen ThermoFisher Scientific, Inc., Los Angeles City, CA, USA).

### 4.8. Cell Invasion and Tumorsphere Assay

Vertical cell motility was evaluated using the matrigel in the Boyden chamber method as previously described by Justus CR, et al. (2014) [44]. The tumor sphere-formation assay was performed according to our previously described method with modifications [17,18]. A small number of colon cancer cells were seeded (2000 cells/well) in six-well ultra-low attachment plates (Corning, Corning, NY, USA) in serum-free media consisting of Dulbecco’s modified Eagle medium (DMEM)/Ham’s F12 (1:1), human epidermal growth factor (hEGF, 20 ng/mL), basic fibroblast growth factor (bFGF; 10 ng/mL (PeproTech, Rocky Hill, NJ, USA), 2 μg/mL 0.2% heparin (Sigma, St. Louis, MO, USA), and 1% penicillin/streptomycin (P/S, 100 U/mL, Hyclone, Logan, UT, USA). Cells were then allowed to grow and make spheres by aggregating for a week, Cells (diameter > 50 µm), characterized by compact, non-adherent spheroid-like masses, were considered a tumor-sphere and counted using an inverted phase-contrast microscope.

### 4.9. Colony Formation Assay

The colony formation assay was performed according to a previously explained protocol [45] with modifications. Briefly, a total of 500 colon cancer cells were seeded in six-well plates. The cells were allowed to grow for another week and then harvested, fixed, and counted.

### 4.10. Detection of Apoptosis through Flow Cytometry

CRC cells were seeded into six-well plates, cultured in DMEM-F12 supplemented with 10% fetal bovine serum (FBS), and incubated at 37 °C in 5% CO_2_ for 24 h, after respective treatments. Cell apoptosis was assayed by using a PE Annexin V Apoptosis Detection Kit I (BD Biosciences) according to the manufacturer’s instructions. Cells were trypsinized using 0.25% trypsin-EDTA solution, washed twice with cold PBS, and stained with Annexin V-PE (5 μL) and 7-AAD (5 μL) in binding buffer. After incubation at room temperature for 15 min, cell apoptosis was analyzed using BD FACS Aria III flow cytometer.

### 4.11. In Vivo Studies

For the in vivo experiments, female BALB/c athymic nude mice (4 weeks old) were purchased from BioLASCO Taiwan Co., Ltd. All the animal experiments and maintenance conformed to the strict compliance of the Animal Use Protocol Taipei Medical University (LAC-2020-0535). A suspension of SW1463 cells (3 × 10^5^ cells per mice) were subcutaneously injected into the left flanks of mice aged 4–6 weeks old (*n* = 8 per group). The mice were maintained under pathogen-free conditions and were provided with sterilized food and water. Tumor formation was monitored every 3 days by using a vernier caliper, and tumor volume was calculated using a modified ellipsoidal formula: ½ length × width^2^. When the mean tumor was palpable (approximately 100 mm^3^), the mice were randomly grouped as follows: treatment with vehicle (saline), 4-AAQB (5.0 mg/kg i.p. thrice a week), cetuximab (10 mg/kg i.p. twice a week), and a combination of 4-AAQB and cetuximab (4-AAQB 5.0 mg/kg i.p. thrice a week and cetuximab 10 mg/kg/ i.p. twice a week) or vehicle as a control. The body weight, tumor volume, and general condition of mice were measured every 3 days. The mice were humanely euthanized when the tumor volume was 2000 mm^3^ or when the tumor size became excessively ulcerated according to the ethical committee guidelines. The tumor tissue was harvested and formalin-fixed for IHC studies. All tumor orthotopic xenograft animal studies were performed in accordance with protocols approved by the Institutional Animal Care and Use Committee of TMU (LAC-2020-0535).

### 4.12. Immunostaining

After the mice were euthanized, the tumor was excised and fixed in formalin, and embedded in paraffin, cut into 5-µm sections, deparaffinized with xylene, dehydrated with graded alcohol, and incubated with warm deionized water containing 0.3% H_2_O_2_ for 30 min. After eliminating endogenous peroxide from the sections, they were blocked with serum and primary antibodies were added to the section with incubation at 4 °C overnight. The next day, the sections were incubated with IgG antibody-HRP followed by dropwise addition of a mixture prepared using biotin and ABC kit for culture. Color was developed using DAB for 10 min. Thereafter, the sections were counterstained with hematoxylin, washed, dehydrated, and permeabilized. Lastly, sections were observed using an optical microscope.

### 4.13. Statistical Analysis

All values reported are the means ± standard deviation, and the data were analyzed with SPSS version 18.0 (SPSS Inc., Chicago, IL, USA). In addition, GraphPad Prism 8.0 software was also used for analysis (San Diego, CA, USA). Student’s t-test was used for comparing between variables. Kaplan–Meier (K-M) survival curves were generated using online cBioPortal tools for TCGA data. *p* < 0.05 was considered statistically significant.

## 5. Conclusions

In conclusion, as shown in the graphical abstract of Figure 7, our study provides evidence to support the therapeutic role of 4-AAQB. Both in vivo and in vitro, 4-AAQB significantly targets KRAS-mutated CRC through the induction of miR-193a-3p expression, thus targeting KRAS-mutant CRC tumorigenesis. Further investigation of 4-AAQB is in progress to understand the complete mechanism and to develop 4-AAQB as a therapeutic agent.

## Figures and Tables

**Figure 1 ijms-22-07508-f001:**
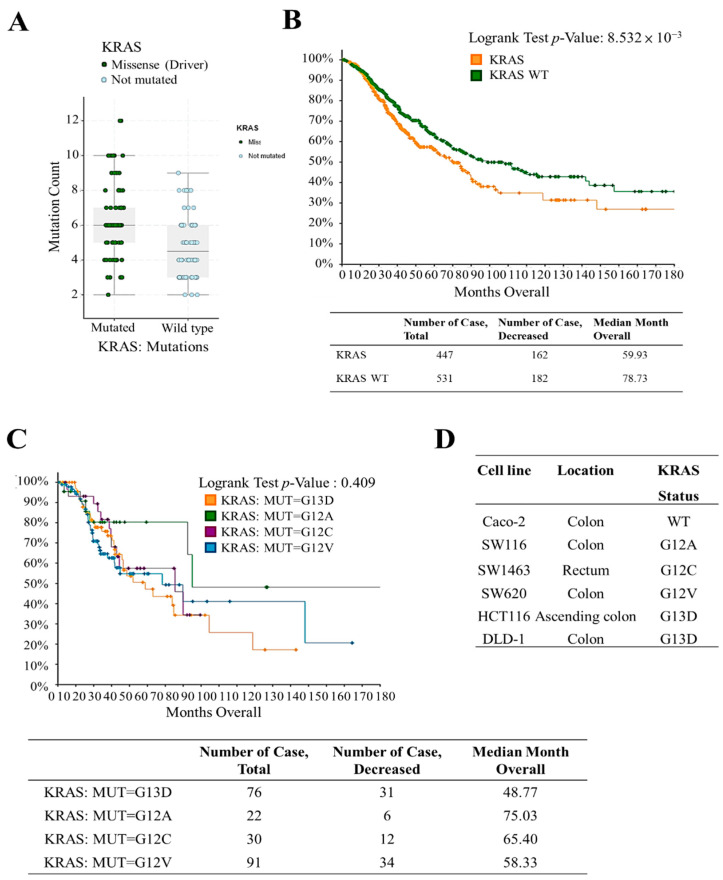
Prevalence of KRAS mutation in patients with colorectal cancer (CRC). (**A**) KRAS mutation count of The Cancer Genome Atlas (TCGA) metastatic colorectal cancer (MSKCC) cohort (MSKCC 2018, *n* = 1134). (**B**) Kaplan–Meier plot showing the overall survival of MSKCC patients according to the genetic profile (KRAS mutation vs. KRAS wild type). (**C**) Overall survival in different subgroups of KRAS mutation (G12A, G12C, G12V, and G13D) obtained by using the same MSKCC colorectal cancer cohort. (**D**) Tissue origin and KRAS status of CRC line recruited in this study. The source of the cell’s information was acquired from SIB Bioinformatics Resource Portal.

**Figure 2 ijms-22-07508-f002:**
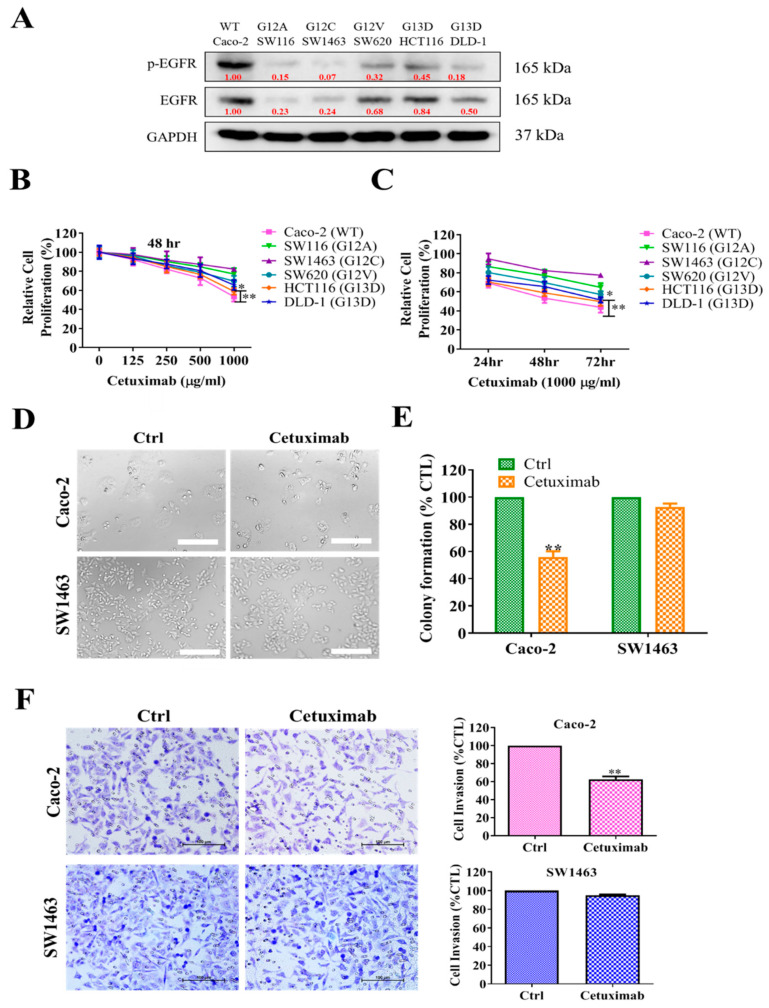
KRAS-mutated CRC cells resist cetuximab treatment. (**A**) Western blot analysis of expression of p-EGFR/EGFR in CRC cells. (**B**,**C**) Cell proliferation analysis by using sulforhodamine B (SRB) assay of KRAS-wild (Caco-2) and KRAS-mutant CRC cell lines. The graphic represents the dose-dependent and time-dependent effect of cetuximab on the relative cell proliferation of CRC cells in relation to controls. (**D**) Effect of cetuximab on CRC cell morphology as obtained through phase-contrast microscopy image at 200 × magnification. (**E**,**F**) Colony formation assay and invasion assay of cetuximab treated and non-treated (control) cells. Representative figures and quantification of the covered areas by using ImageJ are provided in D and F, respectively. * *p* < 0.05, ** *p* < 0.01; scale bar 100 μM.

**Figure 3 ijms-22-07508-f003:**
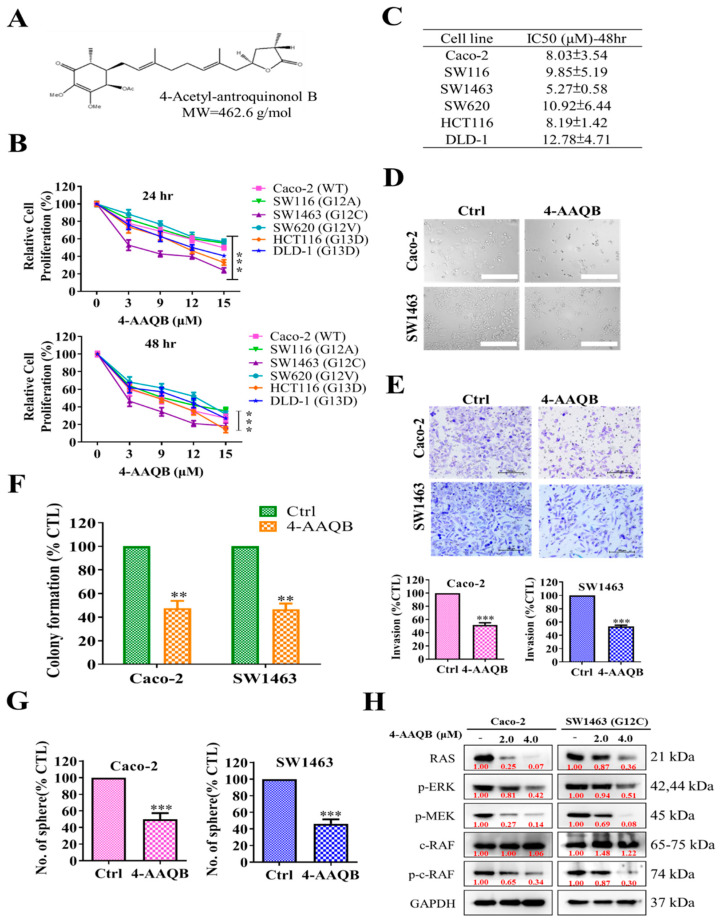
The effect of 4-acetyl-antroquinonol B (4-AAQB) on cell growth and oncogenic characteristics of KRAS wild and KRAS mutant CRC cell lines. (**A**) The chemical structure of 4-AAQB (C26H38O7), (**B**) cell proliferation analysis based on SRB assay of CRC KRAS wild and mutant cell lines. The graphic represents the relative proliferation of the cells following 24 and 48 h of treatment related to control. (**C**) IC_50_-dosage of 4-AAQB on CRC cells. (**D**) Effect of 4-AAQB on the morphologies of the KRAS-mutant (SW1463) and KRAS wild-type (Caco-2) CRC cells as observed through phase-contrast microscopy image at 200× magnification. (**E**–**G**) Reduced migration, colony-forming and tumorsphere formation ability of SW1463 and Caco-2 cells after 4-AAQB treatment. (**H**) Representative image of Western blot analysis was performed to determine the level of key members of the Raf/MEK/ERK and Ras pathway in response to 4-AAQB treatment on the cetuximab resistant CRC KRAS mutant cells. ** *p* < 0.01; *** *p* < 0.001. Scale bar 100 μM.

**Figure 4 ijms-22-07508-f004:**
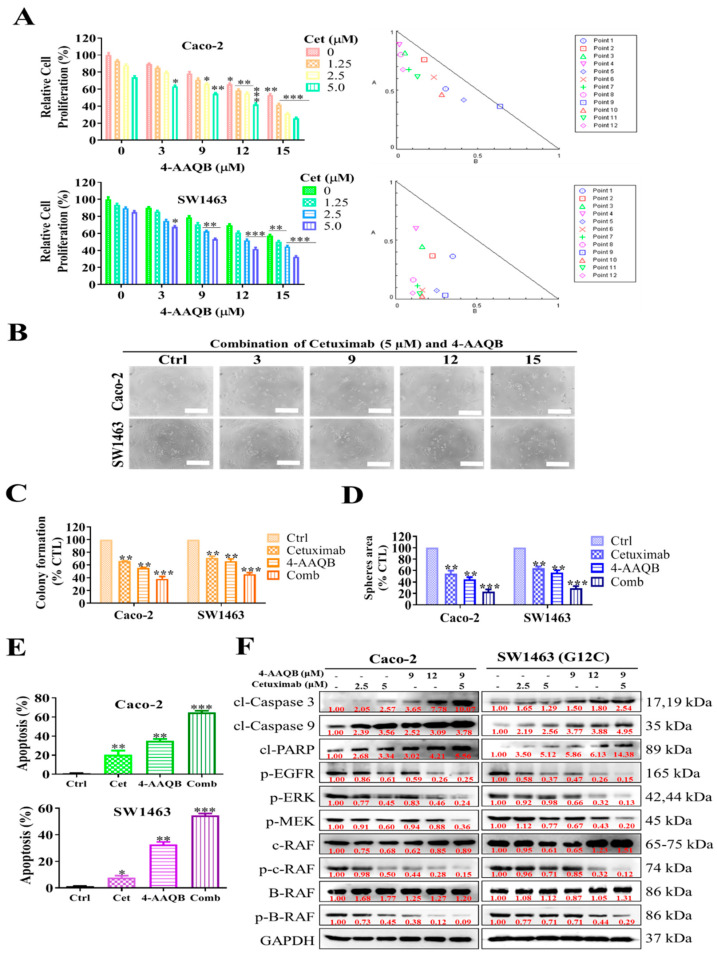
The effect of 4-AAQB and cetuximab combined treatment effect on KRAS wild-type and mutant CRC cell lines in vitro. (**A**) Isobologram analysis showing that the synergistic effects of 4-AAQB and cetuximab were achieved in different concentration combinations on both SW1463 and Caco-2 cell growth. The combination index (CI) using CompuSyn software [26] indicated the synergistic effect of the 4-AAQB-cetuximab combination therapy. (CI > 1.3: antagonism; CI = 1.1–1.3: moderate antagonism; CI = 0.9–1.1: additive effect; CI = 0.8–0.9: slight synergism; CI = 0.6–0.8: moderate synergism; CI = 0.4–0.6: synergism; and CI = 0.2–0.4: strong synergism). (**B**) Effects of 4-AAQB and cetuximab on SW1463 and Caco-2 cell morphology. The phase-contrast microscopy images represent the results from 1 of 3 independent experiments at 200 × magnification. (**C**,**D**) Significant reduction in the colony-forming, and tumorsphere generating abilities, and (**E**) induced apoptosis in SW1463 and Caco-2 cells were observed with the combination treatment. (**F**) Western blot analysis for cl-capase-3/9 and cl-PARP expression, together with the expression of p-EGFR, p-ERK, p-MEK, cRAF/BRAF and p-cRAF/BRAF after the 4-AAQB-cetuximab combination therapy, GAPDH was used as internal housekeeping control. * *p* < 0.05; ** *p* < 0.01; *** *p* < 0.001. Scale bar 100 μM.

**Figure 5 ijms-22-07508-f005:**
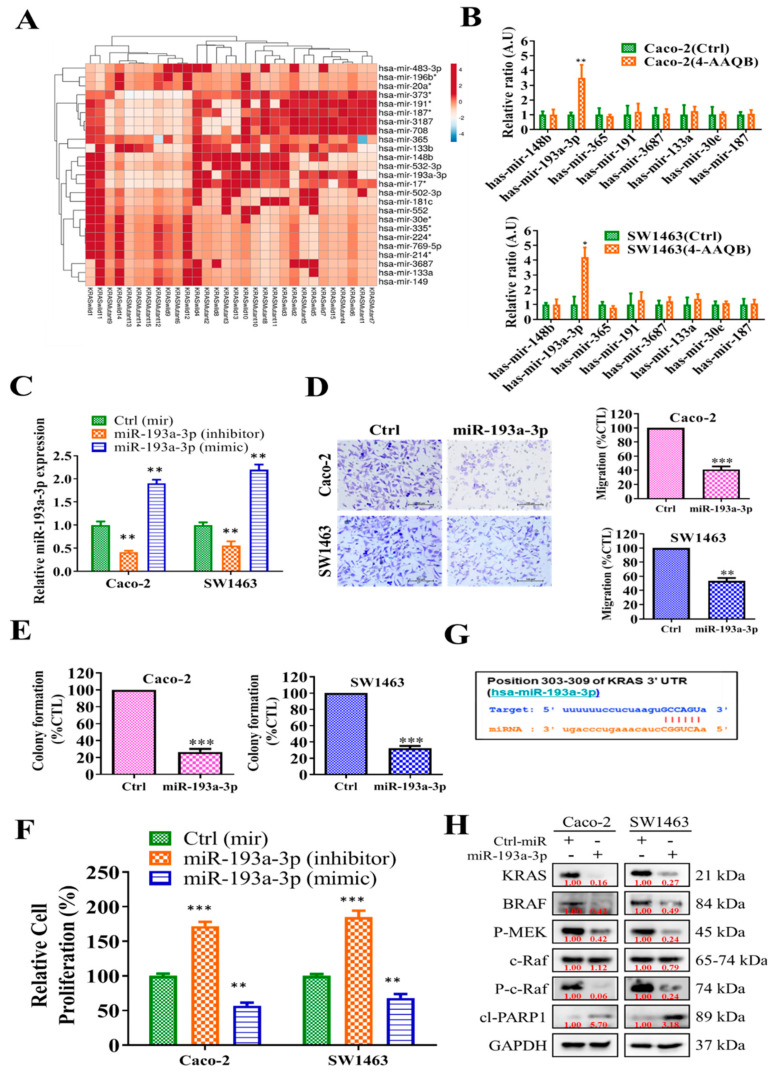
miR-193a-3p impairs the growth and tumorigenicity of KRAS wild-type and KRAS mutant CRC cells. (**A**) Heatmap showing the 25 most downregulated miRs in clinical samples of KRAS-mutated cells in comparison with KRAS wild-type cells downloaded from GEO (GSE66548). (**B**) The quantitative reverse transcription-polymerase chain reaction analysis showed that the 4-AAQB treatment significantly induced the expression of miR-193a-3p in both KRAS mutant and wild-type CRC cells. (**C**) Overexpression and inhibition of miR-193a-3p in CRC cells. (**D**,**E**) Transwell assay was performed to assess the migration ability and colony formation ability of Caco-2 and SW1463 cells after transfection with miR-193a-3p (mimic). (**F**) CRC cells proliferation after miR-193a-3p (mimic or inhibitor) transfection was detected based on SRB assay. (**G**) Bioinformatics analysis indicated that KRAS is predicted as a target gene of miR-193a-3p. (**H**) Western blot analysis of KRAS/BRAF, p-MEK, MEK, p-ERK, ERK, craft, and PAPR1 expression normalized by GAPDH housekeeping gene in the indicated CRC cells. * *p* < 0.05; ** *p* < 0.01; *** *p* < 0.001. Scale bar 100 μM.

**Figure 6 ijms-22-07508-f006:**
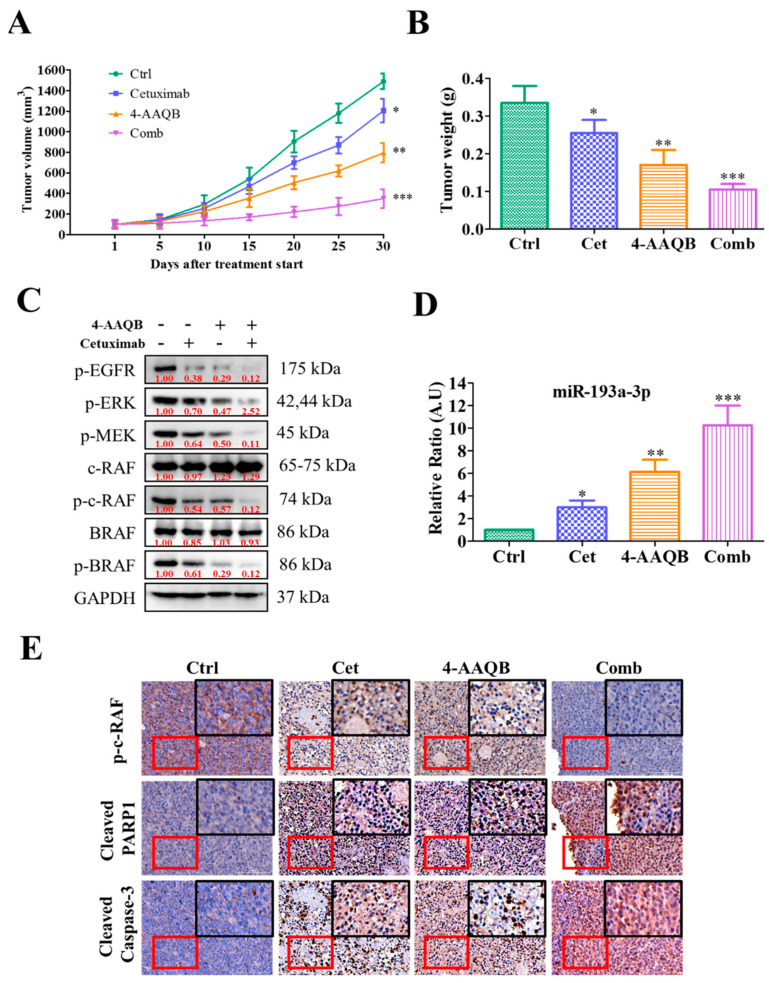
Efficacy evaluation of 4-AAQB by using a KRAS-mutant SW1463 xenograft mouse model. SW1463 tumor treated with the control, cetuximab, 4-AAQB, and combination of 4-AAQB and cetuximab. (**A**) Tumor volume (mm^3^), (**B**) tumor weight (g), (**C**) Western blot analysis of the tumor sample. The expression of p-EGFR, p-ERK, c-RAF/BRAF, and p-c-RAF/BRAF were significantly downregulated in tumor treated with combination and alone 4-AAQB. (**D**) qPCR analyses of the plasma levels of miR-193a-3p. Pooled blood samples of all four groups of mice were analyzed for miR-193a-3p plasma levels. (**E**) Representative IHC staining of p-c-RAF, cl-Caspase-3 and cl-PARP in SW1463 xenografts treated with control, 4-AAQB, cetuximab, or cetuximab combined with 4-AAQB. * *p* < 0.05; ** *p* < 0.01; *** *p* < 0.001.

**Figure 7 ijms-22-07508-f007:**
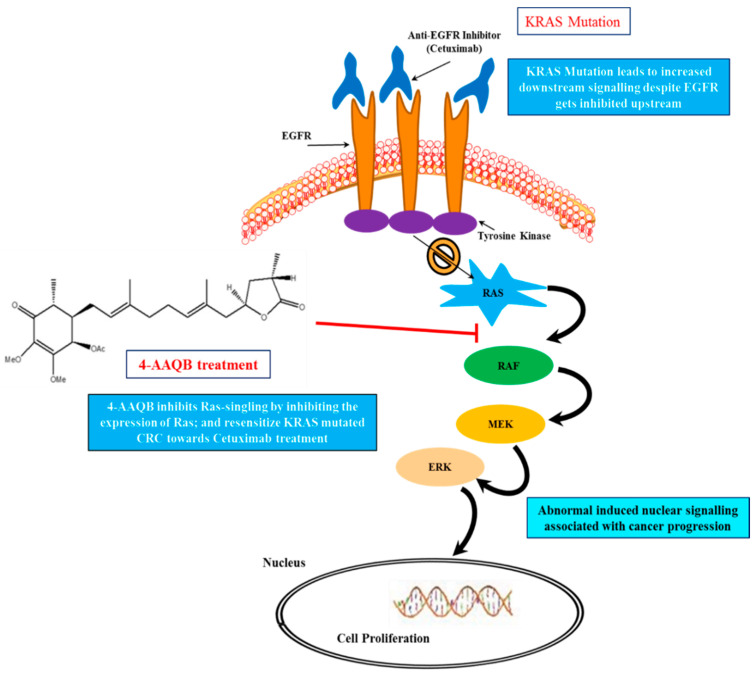
Graphical Abstract: Schematic summary of 4-AAQB inhibiting the Ras signaling pathway and enhancing the anti-EGFR response in KRAS-mutant CRC cells.

## Data Availability

The datasets used and analyzed in the current study are publicly accessible, as indicated in the manuscript.

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
