# Peer review of "4-Acetyl-Antroquinonol B Improves the Sensitization of Cetuximab on Both Kras Mutant and Wild Type Colorectal Cancer by Modulating the Expression of Ras/Raf/miR-193a-3p Signaling Axis"

_ijms, 2021, doi:10.3390/ijms22147508_

Round 1
Reviewer 1 Report
Comments to the authors
The article with the title “4-Acetyl-Antroquinonol B improves the sensitization of Cetuximab on both KRAS mutant and wild type colorectal cancer by modulating the expression of Ras/Raf/miR-193a-3p signalling axis” is in generally well done, but I would offer these comments to the investigators:
- Some minor grammatical errors occur. The manuscript contains significant language-related issues. Please correct these types of grammatical errors throughout the paper.
- The authors treated several CRC cell lines with anti-EGFR MoAb (Cetuximab). I strongly suggest to detect the protein levels of p-EGFR/EGFR in all cell lines (not only Caco-2 and SW1463).
- Figure3: the authors investigated the role of 4-AAQB in RAF/MEK/ERK signaling pathway. Please explain why you used CRAF and not BRAF. It is well known that the C-RAF/B-RAF dimerazation has already been identified as a resistance mechanism against several anti-tumor agents. For the sake of the experiments please identified also the protein levels of BRAF.
- Figure 3: KRAS regulates both MEK/ERK and AKT/mTOR axis. In addition HCT116 and DLD-1 harboring also PI3KCA mutation (H1047R and E545K respectively). I highly recommend to identify the protein levels of AKT/mTOR pathways.
- Figure 4: Authors must detect also the cl. PARP-1. Caspase-3 isprimarily responsible for the cleavage of PARP during apoptotic cell death.
- In in vivo experiments how the authors explain the reduction of tumor volume? Did they measure apoptotic markers as they were identified in in vitro experiments?

Author Response
Response to Reviewers:
Reviewer #1 (Comments to the Author):
1) Some minor grammatical errors occur. The manuscript contains significant language-related issues. Please correct these types of grammatical errors throughout the paper.
A1: Thank you very much for your valuable comments. We sincerely apologize for these errors, and we have revised our manuscript accordingly; we also have taken the help of "Wallace Academic Editing" for professional English editing provided by our institution. https://www.editing.tw/en/brief-introduction-wallace as per the reviewer’s kind suggestions.
2) The authors treated several CRC cells lines with anti-EGFR MoAb (Cetuximab). I strongly suggest detecting the protein levels of p-EGFR/EGFR in all cell lines (not only Caco-2 and SW1463).
A2: We sincerely appreciate reviewers’ insightful comments and suggestions for revising the manuscript. We have now revised our manuscript accordingly, as per the suggestions we detected the expression of p-EGFR/EGFR in Caco-2, SW116, SW1463, SW620, HCT116, DLD-1 CRC cells. Please kindly see our amended Figure 2A and text on line 276 to 293.
3.2. KRAS-Mutated Colorectal Cells Were Resistant to Cetuximab Treatment
KRAS-mutated CRC cells are frequently resistant to anti-EGFR treatments. Protein level of the p-EGFR/EGFR in KRAS-wild (Caco-2) and KRAS-mutant CRC cell lines (Fig. 2A). The dose and time-dependent effect of Cetuximab on cell viability were assessed through the SRB assay. As shown in Fig. 2B, C, KRAS-mutated SW1463 (G12C) cells showed higher resistance to cetuximab in comparison to other KRAS-mutant CRC cells both at dose/time-dependent, whereas maximum inhibition in cell viability of the KRAS-wild type Caco-2 cells was achieved at the given concentration of cetuximab. Complete inhibition of SW1463 cell viability could not be reached with 1000 µg/ml of cetuximab. Thus, anti-EGFR antibody treatment was effective for KRAS-wild Caco-2 cells, but single-drug therapy of cetuximab was not sufficient to achieve a therapeutic outcome on KRAS-mutant CRC cells. Cetuximab treatment did not affect the morphology of the SW1463 cells (cell shrinkage, cytoplasmic membrane blebbing, and cell death) compared with the control untreated cells (Fig. 2D). However, Caco-2 cells were sensitive to cetuximab. We next performed a clonogenic and invasion assay with the indicated treatments. In this assay, cetuximab treatment for Caco-2 cells displayed a 50%-60 % inhibition in colony-forming and invasive properties (Fig. 2E, F). By contrast, no change in the clonogenic and invasive property was noticed for SW1463 cells (Fig. 2E, F).”
Please also see newly Figure 2 legends in line 295 to 303
Figure 2. KRAS-mutated CRC cells resist Cetuximab treatment. (A) Western blot analysis of expression of p-EGFR/EGFR in CRC cells. (B, C) Cell proliferation analysis by using sulforhodamine B (SRB) assay of KRAS-wild (Caco-2) and KRAS-mutant CRC cell lines. The graphic represents the dose-dependent and time-dependent effect of Cetuximab on the relative viability of CRC cells in relation to controls. (D) Effect of Cetuximab on CRC cell morphology as obtained through phase-contrast microscopy image at 200x magnification. (E, F) Colony formation assay and invasion assay of Cetuximab treated and non-treated (control) cells. Representative figures and quantification of the covered areas by using ImageJ are provided in D and F, respectively. * p < 0.05, ** p < 0.01; scale bar 100μm.
3) Figure3: the authors investigated the role of 4-AAQB in RAF/MEK/ERK signaling pathway. Please explain why you used CRAF and not BRAF. It is well known that the C-RAF/B-RAF dimerazation has already been identified as a resistance mechanism against several anti-tumor agents. For the sake of the experiments please identified also the protein levels of BRAF.
A3: We thank the Reviewer for bringing up this good point, and we agree with this. In this revised manuscript, we now have checked the expression B-RAF also. Please kindly see our newly incorporated Supplementary Figure S1 and text on page 9, lines 329-331.
3.3. 4-AAQB Treatment Results in Inhibition of Oncogenic Properties of KRAS-Mutant and Wild-Type Colorectal Cells
We determined whether 4-AAQB (Fig. 3A) treatment could overcome the therapy-resistance of KRAS-mutant CRC cells. A cell viability study of CRC cell lines based on the effect of 4-AAQB treatment was conducted using the SRB assay. A dose-dependent effect was observed at both time points that is, 24 and 48-hours of treatment (Fig. 3B). After 4-AAQB treatment for 48-hours, 50% inhibition of cell viability occurred for all CRC cells within approximately IC50 of 15 μM (Fig. 3C). Furthermore, SW1463 and Caco-2 cell morphology examination under the microscope showed that 4-AAQB treatment effectively induced apoptosis in both cells (Fig. 3D). Next, the effect of 4-AAQB on the invasion and colony-forming abilities of CRC cells (SW1463 and Caco-2) was investigated. Under IC25 treatment with 4-AAQB for 48 hours, the invasive (Fig. 3E) and colony-forming (Fig. 3F) abilities of the cells were greatly inhibited, indicating that 4-AAQB effectively reduced the mobility and invasiveness of the CRC cells, compared with their untreated control counterparts. Furthermore, for determining the effect of 4-AAQB on CRC cell tumorigenesis, we assayed the colon-sphere formation of CRC cells (SW1463 and Caco-2). Tumor-sphere formation assay is crucial for the identification of stemness and drug resistance [26,27]. The tumorsphere formation abilities of SW1463 and Caco-2 were effectively suppressed by 4-AAQB (Fig. 3G). Treatment of SW1463 and Caco-2 cells with 4-AAQB resulted in the reduction of RAF/MEK/ERK signaling and Ras cascade, as shown by a decrease in the protein levels of p-MEK, p-ERK, c-RAF, p-C-RAF, and RAS (Fig. 3H). Furthermore, as shown in Supplementary Fig. S1, reduction in the oncogenic marker expression (p-mTOR, p-AKT, p-PI3K), and B-RAF/p-B-RAF was observed after the 4-AAQB treatment on both the CRC cells (SW1463 and Caco-2). Thus, 4-AAQB plays a pivotal role in the sensitization of KRAS-mutant SW1463 cells to the therapy.
Please also see newly Figure 3 legends in line 331 to 341
Figure 3. The effect of 4-acetyl-antroquinonol B (4-AAQB) on cell growth and oncogenic characteristics of KRAS wild and KRAS mutant CRC cell lines. (A) The chemical structure of 4-AAQB (C26H38O7), (B) Cell proliferation analysis based on SRB assay of CRC KRAS wild and mutant cell lines. The graphic represents the relative proliferation/viability of the cells following 24 and 48 hours of treatment related to control. (C) IC50-Dosage of 4-AAQB on CRC cells. (D) Effect of 4-AAQB on the morphologies of the KRAS-mutant (SW1463) and KRAS wild-type (Caco-2) CRC cells as observed through phase-contrast microscopy image at 200x magnification. (E, G) Reduced migration, colony-forming and tumorsphere formation ability of SW1463 and Caco-2 cells after 4-AAQB treatment. (H) Representative image of western blot analysis was performed to determine the level of key members of the Raf/MEK/ERK and Ras pathway in response to 4-AAQB treatment on the Cetuximab resistant CRC KRAS mutant cells. ** p < 0.01; *** p < 0.001. scale bar 100μm.
Supplementary Figure S1. Reduction in the oncogenic marker expression (p-mTOR, p-AKT, p-PI3K), and B-RAF/p-B-RAF was observed after the 4-AAQB treatment on both the CRC cells (SW1463 and Caco-2).
4) Figure 3: KRAS regulates both MEK/ERK and AKT/mTOR axis. In addition, HCT116 and DLD-1 harboring also PI3KCA mutation (H1047R and E545K respectively). I highly recommend identifying the protein levels of AKT/mTOR pathways.
A4: We are grateful for the reviewer’s comment. Yes, we agree with the reviewer's view. In this revised manuscript, we now have also re-checked and reported the expression of AKT/mTOR. Please kindly see our newly incorporated Supplementary Figure S1 and text on page 9, lines 329-331.
3.3. 4-AAQB Treatment Results in Inhibition of Oncogenic Properties of KRAS-Mutant and Wild-Type Colorectal Cells
We determined whether 4-AAQB (Fig. 3A) treatment could overcome the therapy-resistance of KRAS-mutant CRC cells. A cell viability study of CRC cell lines based on the effect of 4-AAQB treatment was conducted using the SRB assay. A dose-dependent effect was observed at both time points that is, 24 and 48-hours of treatment (Fig. 3B). After 4-AAQB treatment for 48-hours, 50% inhibition of cell viability occurred for all CRC cells within approximately IC50 of 15 μM (Fig. 3C). Furthermore, SW1463 and Caco-2 cell morphology examination under the microscope showed that 4-AAQB treatment effectively induced apoptosis in both cells (Fig. 3D). Next, the effect of 4-AAQB on the invasion and colony-forming abilities of CRC cells (SW1463 and Caco-2) was investigated. Under IC25 treatment with 4-AAQB for 48 hours, the invasive (Fig. 3E) and colony-forming (Fig. 3F) abilities of the cells were greatly inhibited, indicating that 4-AAQB effectively reduced the mobility and invasiveness of the CRC cells, compared with their untreated control counterparts. Furthermore, for determining the effect of 4-AAQB on CRC cell tumorigenesis, we assayed the colon-sphere formation of CRC cells (SW1463 and Caco-2). Tumor-sphere formation assay is crucial for the identification of stemness and drug resistance [26,27]. The tumorsphere formation abilities of SW1463 and Caco-2 were effectively suppressed by 4-AAQB (Fig. 3G). Treatment of SW1463 and Caco-2 cells with 4-AAQB resulted in the reduction of RAF/MEK/ERK signaling and Ras cascade, as shown by a decrease in the protein levels of p-MEK, p-ERK, c-RAF, p-C-RAF, and RAS (Fig. 3H). Furthermore, as shown in Supplementary Fig. S1, reduction in the oncogenic marker expression (p-mTOR, p-AKT, p-PI3K), and B-RAF/p-B-RAF was observed after the 4-AAQB treatment on both the CRC cells (SW1463 and Caco-2). Thus, 4-AAQB plays a pivotal role in the sensitization of KRAS-mutant SW1463 cells to the therapy.
Please also see newly Figure 3 legends in line 331 to 341
Figure 3. The effect of 4-acetyl-antroquinonol B (4-AAQB) on cell growth and oncogenic characteristics of KRAS wild and KRAS mutant CRC cell lines. (A) The chemical structure of 4-AAQB (C26H38O7), (B) Cell proliferation analysis based on SRB assay of CRC KRAS wild and mutant cell lines. The graphic represents the relative proliferation/viability of the cells following 24 and 48 hours of treatment related to control. (C) IC50-Dosage of 4-AAQB on CRC cells. (D) Effect of 4-AAQB on the morphologies of the KRAS-mutant (SW1463) and KRAS wild-type (Caco-2) CRC cells as observed through phase-contrast microscopy image at 200x magnification. (E, G) Reduced migration, colony-forming and tumorsphere formation ability of SW1463 and Caco-2 cells after 4-AAQB treatment. (H) Representative image of western blot analysis was performed to determine the level of key members of the Raf/MEK/ERK and Ras pathway in response to 4-AAQB treatment on the Cetuximab resistant CRC KRAS mutant cells. ** p < 0.01; *** p < 0.001. scale bar 100μm.
Supplementary Figure S1. Reduction in the oncogenic marker expression (p-mTOR, p-AKT, p-PI3K), and B-RAF/p-B-RAF was observed after the 4-AAQB treatment on both the CRC cells (SW1463 and Caco-2).
5) Figure 4: Authors must detect also the cl. PARP-1. Caspase-3 is primarily responsible for the cleavage of PARP during apoptotic cell death.
A5: We thank the Reviewer for bringing up this good point, and we agree with this. We checked the expression of cleaved PARP1 through western blot as suggested by the reviewer, we have now revised the result and figure. Please kindly see our revised Figure 4F and result in line 343 to 380.
3.4. Combination Treatment with 4-AAQB and Cetuximab Increased Cetuximab Sensitivity in KRAS-Mutant CRC Cells
Epidermal growth factor receptor (EGFR), a protein tyrosine kinase receptor, is frequently expressed in CRC and is involved in cell proliferation and cell survival [28]. Anti-EGFR therapy including cetuximab and panitumumab significantly improves the survival of KRAS wild-type MSKCC, but not of those with KRAS-mutant cancer [29], suggesting that the single-drug therapy of cetuximab is insufficient to achieve a therapeutic effect. To qualitatively analyze whether the combination of 4-AAQB and cetuximab could produce synergistic anti-proliferative effects, combination index (CI) is calculated. CI values at IC50 points were calculated according to the results of SRB assays and the CI values are shown in Fig. 4A. Treatment with 4-AAQB in combination with cetuximab exhibited an overall synergistic effect (Fig. 4B). In KRAS wild-type Caco-2 cells, low-dose (3-15 μM) 4-AAQB demonstrated a slight synergistic effect, but when the 4-AAQB concentration was 12 μM in combination with 5 μM cetuximab, the CI value increased to 1, which indicates only an additive effect at this concentration. Moreover, increasing the concentration of 4-AAQB to 15 μM decreased the CI value to 0.72–0.75, indicating a moderate synergistic effect of this drug in combination with cetuximab. In KRAS mutant-SW1463 cells, the synergistic effect leads to a re-boost. With low-dose (3 μM) 4-AAQB, the CI value ranges from 0.6 to 0.72, demonstrating moderate synergism of the drug in combination with cetuximab. Furthermore, increasing the concentration of 4-AAQB to 3–15 μM decreased the CI value from 0.24 to 0.34, which indicates strong synergism. In the high-dose 4-AAQB group, the CI value further decreased to <0.2 (0.16–0.19), indicating an excellent synergistic effect. Thus, we also tested whether the combination of 4-AAQB and cetuximab can remarkably suppress CRC colony formation, tumorsphere generation, and cell proliferation abilities. Strikingly, the combination of 4-AAQB and cetuximab treatment synergistically resensitized cells and inhibited CRC colony formation, tumorsphere generation, and cell proliferation abilities through the induction of apoptosis (Fig. 4C–E). To further understand whether the anti-proliferative effects on cell viability involve apoptotic machinery, SW1463 and Caco-2 cells were treated with 9–12 µM 4-AAQB or 2.5–5 µM cetuximab or their combination for 24 hours; then, protein assays were conducted to determine the expression level of cleaved (cl) caspase-3/9 and PARP. Treatment with 4-AAQB and cetuximab alone enhanced the expression of cl-caspase-3 and cl-PARP as compared with the control (Fig. 4F). Combination treatment induced a more than twofold increase in the expression of cl-caspase-3 and cl-PARP in SW1463. These results illustrated that the synergistic anti-proliferative effect of 4-AAQB and cetuximab is through cl-caspase-3/9 and cl-PARP expression on CRC cells. Together with the induction of expression apoptotic markers, combination treatment effectively modulates the expression of activated p-EGFR-B-raf/c-Raf-Erk-Mek, as described in the Western blot image (Fig. 4F).
Please also see newly Figure 4 legends in line 382 to 395
Figure 4. The effect of 4-AAQB and cetuximab combined treatment effect on KRAS wild-type and mutant CRC cell lines in vitro. (A) Isobologram analysis showing that the synergistic effects of 4-AAQB and cetuximab were achieved in different concentration combinations on both SW1463 and Caco-2 cell growth. The combination index (CI) using CompuSyn software[30] indicated the synergistic effect of the 4-AAQB-cetuximab combination therapy. (CI > 1.3: antagonism; CI = 1.1-1.3: moderate antagonism; CI = 0.9-1.1: additive effect; CI = 0.8-0.9: slight synergism; CI = 0.6-0.8: moderate synergism; CI 0.4-0.6: synergism; and CI 0.2-0.4: strong synergism). (B) Effects of 4-AAQB and Cetuximab on SW1463 and Caco-2 cell morphology. The phase-contrast microscopy images represent the results from 1 of 3 independent experiments at 200x magnification. (C, D) Significant reduction in the colony-forming, and tumorsphere generating abilities, and (E) induced apoptosis in SW1463 and Caco-2 cells were observed with the combination treatment. (F) Western blot analysis for cl-capase3/9 and cl-PARP expression, together with the expression of p-EGFR, p-ERK, p-MEK, cRAF/BRAF and p-cRAF/BRAF after the 4-AAQB-cetuximab combination therapy, GAPDH was used as internal housekeeping control. * p < 0.05; ** p < 0.01; *** p < 0.001. scale bar 100μm.
6) In in vivo experiments how the authors explain the reduction of tumor volume? A6: We appreciate the reviewer’s comments. From in-vitro, we observed 4-AAQB alone or in synergism with Cetuximab reduces the tumorigenicity of CRC cells via modulating the expression of EGFR/ERK/MEK/cRaf/BRaf/miR-193a-3p and inducing the apoptosis of CRC cells. Our In-vivo data also demonstrated 4-AAQB alone or in combination with Cetuximab again effectively modulate the expression of EGFR/ERK/MEK/cRaf/BRaf/Capase-3/cl-PARP in tumor tissue samples. Whereas the overexpression of tumor suppressor miRNA-193a-3p and apoptotic markers were observed in mice blood and tumor tissue samples, respectively, this suggests to us the tumour-suppressive role of 4-AAQB, which is reflecting through the tumor volume reduction. Please kindly see our revised Figure 6 and result in line 432 to 447.
3.6. Treatment with 4-AAQB Increased Cetuximab Efficacy In Vivo
After establishing the anti-CRC role of 4-AAQB in vitro, for our in vivo study, we evaluated the effect of 4-AAQB by using a xenograft mouse SW1463 tumor model. The tumor size and volume over time clearly showed that combination treatment with 4-AAQB and cetuximab significantly delayed tumorigenesis followed by 4-AAQB and cetuximab treatment alone compared with vehicle (Fig. 6A and B). Western blot analysis of tumor samples collected from all groups demonstrated the changes in the expres-sions of oncogenic markers (p-EGFR), p-ERK, p-MEK, c-RAF/BRAF and p-c-RAF/BRAF (Fig. 6C), whereas the qRT-PCR analysis of blood-plasma levels showed the highest level of miR-193a-3p expression in combination-treated pooled blood samples, fol-lowed by 4-AAQB, Cetuximab, and vehicle control (Fig. 6D). As expected, IHC analysis of combination treatment indicated that the expression of activating c-RAF protein level was effectively reduced, whereas, the expression of apoptotic markers (p-c-RAF, cl-Caspase3 and cl-PARP) was induced in the combination treatment group as com-pared to alone 4-AAQB and cetuximab treatment and vehicle control (Fig. 6E) resulting in the reduction of tumor burden in the xenograft mouse model.
Please also see newly Figure 6 legends in line 450 to 458
Figure 6. Efficacy evaluation of 4-AAQB by using a KRAS-mutant SW1463 xenograft mouse model. SW1463 tumor treated with the control, cetuximab, 4-AAQB, and combination of 4-AAQB and cetuximab. (A) Tumor volume (mm3), (B) Tumor weight (g), (C) Western blot analysis of the tumor sample. The expression of p-EGFR, p-ERK, c-RAF/BRAF, and p-c-RAF/BRAF were significantly downregulated in tumor treated with combination and alone 4-AAQB. (D) qPCR analyses of the plasma levels of miR-193a-3p. Pooled blood samples of all four groups of mice were analyzed for miR-193a-3p plasma levels. (E) Representative IHC staining of p-c-RAF, cl-Caspase-3 and cl-PARP in SW1463 xenografts treated with control, 4-AAQB, cetuximab, or cetuximab combined with 4-AAQB. * p < 0.05; ** p < 0.01; *** p < 0.001.
7) Did they measure apoptotic markers as they were identified in in vitro experiments?
A7: We are grateful for the reviewer’s comment. We did not measure the expression of apoptotic markers in the in-vivo results. But as per the suggestions we checked the expression of apoptotic (Caspase3 & cleaved PARP1) markers in our IHC and observed induction in the expression of cl-Caspase-3 & cl-PARP1apoptotic markers. Please kindly see our revised Figure 6E and result in line 432 to 447.
3.6. Treatment with 4-AAQB Increased Cetuximab Efficacy In Vivo
After establishing the anti-CRC role of 4-AAQB in vitro, for our in vivo study, we evaluated the effect of 4-AAQB by using a xenograft mouse SW1463 tumor model. The tumor size and volume over time clearly showed that combination treatment with 4-AAQB and cetuximab significantly delayed tumorigenesis followed by 4-AAQB and cetuximab treatment alone compared with vehicle (Fig. 6A and B). Western blot analysis of tumor samples collected from all groups demonstrated the changes in the expres-sions of oncogenic markers (p-EGFR), p-ERK, p-MEK, c-RAF/BRAF and p-c-RAF/BRAF (Fig. 6C), whereas the qRT-PCR analysis of blood-plasma levels showed the highest level of miR-193a-3p expression in combination-treated pooled blood samples, fol-lowed by 4-AAQB, Cetuximab, and vehicle control (Fig. 6D). As expected, IHC analysis of combination treatment indicated that the expression of activating c-RAF protein level was effectively reduced, whereas, the expression of apoptotic markers (p-c-RAF, cl-Caspase3 and cl-PARP) was induced in the combination treatment group as com-pared to alone 4-AAQB and cetuximab treatment and vehicle control (Fig. 6E) resulting in the reduction of tumor burden in the xenograft mouse model.
Please also see newly Figure 6 legends in line 450 to 458
Figure 6. Efficacy evaluation of 4-AAQB by using a KRAS-mutant SW1463 xenograft mouse model. SW1463 tumor treated with the control, cetuximab, 4-AAQB, and combination of 4-AAQB and cetuximab. (A) Tumor volume (mm3), (B) Tumor weight (g), (C) Western blot analysis of the tumor sample. The expression of p-EGFR, p-ERK, c-RAF/BRAF, and p-c-RAF/BRAF were significantly downregulated in tumor treated with combination and alone 4-AAQB. (D) qPCR analyses of the plasma levels of miR-193a-3p. Pooled blood samples of all four groups of mice were analyzed for miR-193a-3p plasma levels. (E) Representative IHC staining of p-c-RAF, cl-Caspase-3 and cl-PARP in SW1463 xenografts treated with control, 4-AAQB, cetuximab, or cetuximab combined with 4-AAQB. * p < 0.05; ** p < 0.01; *** p < 0.001.

Reviewer 2 Report
In this research paper, the authors investigated the effect of 4-AAQB in colorectal cancer cell lines that harbor KRAS mutations and cells that express wild type KRAS protein. Using a variety of assays, they showed that KRAS mutant SW1463 cells were resistant to cetuximab therapy; however, treatment with 4-AAQB resensitized CRC cells towards cetuximab therapy. This was associated with reduced KRAS signaling pathway in CRC cells, and reduced colony forming ability. Such an effect was not observed in cells containing wild KRAS. Further analysis revealed that miR-193a-3p was significantly down-regulated in the KRAS-mutated patient samples compared to KRAS-wild CRC patients. Overexpression of mIR-193a-3p resulted in a significant reduction of oncogenecity. They suggest that KRAs is a target for miR-193-3p. The in vivo treatment with 4-AAQB in combination with Cetuximab reduced the expression of EGFR, pMEK, pERK, c-RAF/p-cRAF signaling as well as induced the expression of plasma miR-193a-3p level. The authors conclude that 4-AAQB inhibit Ras signaling in CRC cell and re-sensitizes treatment with Cetuximab.
Minor comments:
- It is not clear from the paper how treatment with 4-AAQB of KRAS mutant cells results in the over expression of miR-193a-3p. Please explain. Is it entirely through modulation of signaling pathway, or are there other direct protein targets for this drug? There are many reports that describes the inhibitory effect of 4-AAQB in cancer cells. What are the direct targets of 4-AAQB?
- miR-193a-3p acts as a tumor suppressor miRNA by targeting genes involved in cell proliferation, apoptosis, migration, invasion and metastasis. Is anything known in literature on the genes that are modulated by miR-193a-3p? Please include the details in the text.
- A model describing the mechanisms of action of 4-AAQB in resensitizing the cetucximab treatment in cells will be helpful.
Author Response
Response to Reviewers:
Reviewer #2 (Comments to the Author):
1)It is not clear from the paper how treatment with 4-AAQB of KRAS mutant cells results in the over expression of miR-193a-3p. Please explain. Is it entirely through modulation of signaling pathway, or are there other direct protein targets for this drug?
A1: We are grateful to the Reviewer for inviting us to clarify this important point. Please refer to the discussion in our revised manuscript in line 460-539
- Discussion
Cancers of the GI tract are a major cause of mortality in patients, mainly CRC [2]. CRC is the third ubiquitous cancer worldwide and in Asia [3]. Intrinsic and acquired therapy resistance are the main hurdles to overcome for maximizing the benefits of therapy. Notably, the KRAS mutation is associated with low overall survival in pa-tients with advanced-stage CRC who are treated with cetuximab after radiation ther-apy [31]. The use of anti-EGFR therapy including cetuximab and panitumumab had significantly improved the survival of KRAS wild-type MSKCC patients, but for the KRAS mutant group, it seems ineffective [29]. Other studies have hypothesized that the anti-EGFR effect is Ras-dependent [32]. Overexpression of the Ras-signaling pathway in KRAS-mutant CRC disrupts the downstream signal transduction of anti-EGFR therapy that render KRAS-mutant CRC resistant to cetuximab [33]. Therefore, KRAS-mutant CRC is a good research model to study the biology and mechanism of anti-EGFR therapy as well as to establish and discovery of new therapeutic strategies for reverting the resistance of KRAS-mutant CRC to cetuximab.
In this present study, we did in-silico analysis to investigate the expression of miRNA’s in KRAS-mutant vs KRAS-wild CRC samples, and observed miR-193a-3p is significantly suppressed in mutant KRAS samples as compared to control (wild) sam-ples. In-vitro analysis of gain/loss of function of miR-193a-3p in the CRC cells demon-strated its opposite correlation with the expression of KRAS. The converse miR-193a-3p/KRAS relationship is associated with enhanced oncogenicity of CRC cells. Whereas, when 4-AAQB is introduced, it modulates the re-expression of miR-193a-3p, akin to short-interfering RNA, effectively reduced the expression of KRAS, results in suppression of CRC oncogenicity and tumorigenesis. Tumor suppressive effect of miR-193a-3p in cancer is well studied, especially its anti-lung cancer effect via target-ing KRAS. Studies reported and confirmed KRAS is a potential target of miR-193a-3p. Therefore, induction of miR-193a-3p expression via 4-AAQB might modulate the an-ti-CRC effect. Previously our group reported the pivotal 4-AAQB’s anti-CRC role on CRC cell lines through impeding the formation of CRC cancer stem cells and ROS (Re-active oxygen species) oxidative stress, resultant in the modulation of CRC cells innate or acquired insensitivity towards chemotherapy.
In this study, we examined, that the association of KRAS-mutation status with overall survival among CRC patients (Fig. 1) and showed the resistance of KRAS-mutated CRC cells to cetuximab (Fig. 2A). The cell viability assay showed that KRAS-mutant SW1463 cells were cetuximab-resistant in comparison with Caco-2 cells (Fig. 2B). Furthermore, no change in the clonogenic and invasive properties of SW1463 cells was noted after cetuximab treatment, indicating their resistance to cetuximab (Fig. 2D, F).
Drug resistance caused by the overexpression of the Ras-signaling pathway fre-quently occurs in KRAS-mutant CRC [34]. Inhibiting the Ras-signaling is crucial to overcome its drug resistance [35]. Thus, we hypothesized that modulation of the Ras-signaling pathway by using 4-AAQB, could resensitize KRAS-mutant CRC cells to anti-EGFR therapy. Our present work is the first time to suggest that 4-AAQB treat-ment effectively targets KRAS-mutant and wild-type CRC cells and resensitizes them to cetuximab therapy. The 4-AAQB treatment effectively reduced the viability of CRC cells dose-dependently along with the reduction in the oncogenic property and tu-mor-sphere generation abilities of CRC cells (SW1463 and Caco-2) (Fig. 3). Protein ex-pression revealed that 4-AAQB treatment reduced the phosphorylation activation of key targets participating in the Raf/Mek/Erk pathways and RAS cascade (Fig. 3H) [36]. This is consistent with our previous study, which showed that 4-AAQB suppressed tumorigenesis and inhibited cancer stem cell-like phenotype through multiple signal-ing pathways, including JAK-STAT and Wnt//β-catenin [37]. Moreover, we demon-strated that 4-AAQB induced the apoptosis of CRC cells through the suppression of Ras-signaling pathway via modulating the expression of cl-Caspase3/9 and cl-PARP [38]. Furthermore, few studies have reported that increased autophagy is associated with anti-EGFR drug resistance [39]. Our previous study reported the effect of 4-AAQB on ovarian cancer [40]; 4-AAQB could disrupt autophagy through the inhibition of the upstream signal of ATG (autophagy-related gene).
Importantly, for the first time as of our knowledge to demonstrate that through the inhibition of the Ras-signaling pathway, 4-AAQB can inhibit cancer cell growth and proliferation and induces apoptosis. Ras-signaling is involved in anti-EGFR ther-apy because Ras-protein overexpression causes the inhibitory signal from EGFR [41]. For validation, we comparatively assessed the effect of 4-AAQB and cetuximab as sin-gle-drug and in combination in KRAS wild-type Caco-2 cells human CRC cell and KRAS-mutant SW1463 human CRC cells (Fig. 4). Our result demonstrated that the combination treatment with 4-AAQB and cetuximab could synergistically overcome the resistance of KRAS-mutant CRC cells (SW1463) and KRAS-wild (Caco-2) cells to cetuximab. These findings illustrate that 4-AAQB inhibits Ras-signaling through phosphorylation modification of the RAF/MEK/ERK and Ras cascade, enhancing the potency of anti-EGFR therapy. Furthermore, regarding epigenetic factors controlling the severity of KRAS-mutant CRC cells, we found that miR-193a-3p expression was effectively downregulated in patients with KRAS-mutant CRC (Fig. 5). Furthermore, a previous study also suggested that miR-193-3p inhibition in CRC patients regulated tumor progression [42]. Conversely, miR-193a-3p overexpression inhibited tumor pro-gression. Both in-vitro and in-vivo, 4-AAQB alone or in combination with cetuximab increased miR-193a-3p expression in the xenograft mice model (Fig. 5 and 6).
As EGFR is the upstream initiator of the Ras pathway, inhibiting EGFR cannot ef-fectively inhibit downstream signals when Ras drives the whole pathway [43]. Resen-sitization of KRAS-mutant CRC to cetuximab after turning off Ras signaling has been reported in several studies [44]. 4-AAQB which shares its structure to antroquinonol (AQ), has demonstrated good anticancer potency in CRC [45]. Both 4-AAQB and AQ suppresses Ras-signaling, but which drug has a stronger effect on CRC remains elusive. We hope that use of the natural compounds of 4-AAQB would benefit many patients with CRC harboring KRAS mutations.
2)There are many reports that describes the inhibitory effect of 4-AAQB in cancer cells. What are the direct targets of 4-AAQB?
A2: We are grateful for the reviewer's comment. Yes, from literature 4-AAQB targets many key genes involved in stemness of cancer or drug resistance and autophagy associated, such as LGR5, MTOR, NANOG, SOX2, STAT3, JAK2, ATG5, ATG7, CTNNB1.
3)miR-193a-3p acts as a tumor suppressor miRNA by targeting genes involved in cell proliferation, apoptosis, migration, invasion and metastasis. Is anything known in literature on the genes that are modulated by miR-193a-3p? Please include the details in the text.
A3: We are grateful for the reviewer's comment. miR-193a-3p modulates the expression of KRAS, CCND1, PLAU, and ERBB4 and many important genes. As per suggestions we have incorporated the details of miR-193a-3p target genes into the main text. Please kindly see the introduction section of the main text in line 54-402.
- Introduction
Cancer is one of the major causes of death worldwide [1]. Among organ cancers, gastrointestinal (GI) tract cancer is the major cause of mortality in patients, particu-larly colorectal cancer (CRC) [2]. CRC is the third ubiquitous cancer worldwide and in Asia [3]. For a long time, sporadic CRC has been perceived as a homogenous condition that occurs with the adenoma-carcinoma phenotype, which is caused by many genetic al-terations of CRC key genes, such as KRAS, tumor suppressor protein (TP)53, and APC Regulator Of WNT Signaling Pathway (APC) [4]. However, with breakthrough genetic technologies providing a clear understanding of the driver mutation effect in different cancer types, we are entering the era of precision medicine [5]. KRAS-mutant CRC is associated with poor prognosis (high rate of metastasis and incidence of therapy re-sistance) [6].
As the KRAS mutation is the major driver mutation in CRC [7], developing a drug targeting the Ras pathway is urgently needed. Recently, a promising new drug called Sotorasib (AMG 510), which is a novel KRAS inhibitor targeting the KRAS G12C mu-tation, was approved for treating patients with metastatic non-small-cell lung cancer harboring the KRAS G12C mutation [8]. This illustrates that developing Ras inhibitors for treating cancer is important, and it might become the next step in cancer treatment.
The mutation status of the KRAS gene has been observed to affect the response of CRC toward cetuximab treatment, cetuximab a monoclonal antibody that binds to the extracellular region of EGFR, is effective on KRAS wild type metastatic CRC [9-11]. This drug demonstrates anti-cancer effects through the suppression of EGFRs; it was developed in the 1970s and plays a crucial role in metastatic colorectal cancer [12]. However, patients with KRAS-mutant CRC do not benefit from this treatment[9]. Sev-eral studies have illustrated that Ras signaling overexpression can compensate for EGFR inhibition, causing the failure of anti-EGFR treatment in KRAS-mutant CRC [13]. Thus, the blockage of the Ras signal pathway can resensitize KRAS-mutant CRC cell lines to anti-EGFR effect treatment.
Antrodia cinnamomea is a unique fungus, which is exclusively found in Taiwan. It is traditionally known to have anti-cancer properties. In a broad spectrum of cancers, 4-acetyl-antroquinonol B (4-AAQB) isolated and purified from A. cinnamomea exerts anti-proliferative effects [14-16]. Our previous study illustrated the anti-CRC role of 4-AAQB, which is mediated through the inhibition of the formation of CRC cancer stem cells and reactive oxygen species (ROS) oxidative stress, resulting in the modula-tion of CRC cells’ innate or acquired insensitivity towards chemotherapy [17,18]. Growing evidence exists of the role of small non-coding RNA, particularly micro-RNA (miRNA’s), in controlling the key biological process, including deciding the fate of cancer treatment [19]. Such as the tumour-suppressive effect of miR-193a-3p in lung cancer through targeting KRAS expression [20,21]. As the sequel of our previous pub-lished work, in this study, we specifically targeted CRC cells harboring the KRAS mu-tation and the results showed that KRAS-mutant CRC cells are adequately resistant to cetuximab (anti-EGFR monoclonal antibodies). In-vitro studies of 4-AAQB alone, or in combination have demonstrated significant anti-cancer effects on KRAS mutant CRC cell lines. Our study results, both in-silico and in-vitro, suggested that the miR-193a-3p expression could predict the potential response of KRAS-mutant CRC cells to the treatment. Targeting KRAS -mutant CRC cells and mice xenograft model with 4-AAQB results in the over-expression of miR-193a-3p and the reduction of CRC tumorigenesis. In summary, both in-vivo and in-vitro studies indicate that 4-AAQB may be an im-portant therapeutic agent that targets KRAS-mutant CRC cells through the reduction of the Ras-signaling cascade and modulation of the expression key miR’s in CRC tu-morigenesis.
4) A model describing the mechanisms of action of 4-AAQB in resensitizing the cetucximab treatment in cells will be helpful.
A4: We sincerely thank the reviewer for the time taken to review our work, and for the suggestions given. As per the suggestions we have drawn a model to show describe the mechanism of action of 4-AAQB in resensitizing the Cetuximab via KRAS. Please kindly see our new attached Graphical Abstract: Schematic summary Figure 7, on line 540-550.
- Conclusion
In conclusion, as shown in the graphical abstract of Fig. 7, our study provides ev-idence to support the therapeutic role of 4-AAQB. Both in-vivo and in-vitro, 4-AAQB significantly targets KRAS-mutated CRC through the induction of miR-193a-3p ex-pression, thus targeting KRAS-mutant CRC tumorigenesis. Further investigation of 4-AAQB is in progression to understand the complete mechanism and to develop 4-AAQB as a therapeutic agent.”
Figure 7. Graphical Abstract: Schematic summary of 4-AAQB inhibiting the Ras signaling path-way and enhancing the anti-EGFR response in KRAS-mutant CRC cells.”

Reviewer 3 Report
The manuscript investigates the ability of 4-Acetyl-Antroquinonol B, isolated and purified from Antrodia cinnamomea, a fungus exclusively found in Taiwan, in sensitizing KRAS and wild type colorectal cancer cells to Cetuximab.
The topic could be interesting, but major critical points are present.
- In the manuscript several and serious mistakes in English style and grammar are present; this makes very difficult to read the manuscript and understand the results. A very careful review has to be done.
- In the Results sections 3.1, results regarding the screening of a patient cohort is reported, but in Materials & Methods no description of this activity is present. In a similar way, in 3.5 section Fig. 5A shows an heatmap regarding the expression of miR’s, but no information about this analysis is provided in Materials & Methods.
- The Discussion has to be more focused on the results of the research, avoiding the repetition of concepts already reported in the Introduction.
- Since the authors affirm that Antrodia Cinnamomea is present only in Taiwan, they have to discuss the possibility to synthetize 4-Acetyl-Antroquinonol B or to grow the fungus in other countries. Otherwise, the impact of their results is significantly reduced.
•INTRODUCTION
The sentence “One of the classic examples of KRAS mutation causing drug resistance in colorectal cancer is Cetuximab, an anti-EGFR targeting monoclonal antibody” is wrong because Cetuximab is not a KRAS mutation.
- MATERIALS & METHODS.
- The section 2.2 can be included in 2.1, since it refers to specific products.
- Section 2.5. More information about cell detachment has to be added.
- Since Section 2.10 refers to an “in vivo” analysis, it must to be placed after 2.11.
- The effect on apoptosis has been evaluated by determining caspase 3 and 9 in Western Blot. The authors have to specify if they analysed pro-caspases or cleaved caspases that are the active form of these enzymes.
- FIGURES AND RESULTS.
Figure 3b. The effect of 4-AAQB seems to be dose-dependent, but no information is reported with regard to the time-dependency.
Figure 4f. Why in the combination treatment 9 µM 4-AAQB has been used instead of 12 µM?
Figure 5b. Statistical analysis is not reported even if in the text it is affirmed that the expression of 25 miR’s is significantly lower in KRAS mutant CRC patients.
Author Response
Response to Reviewers:
Reviewer #3 (Comments to the Author):
1)In the manuscript several and serious mistakes in English style and grammar are present; this makes very difficult to read the manuscript and understand the results. A very careful review has to be done.
A1: We sincerely thank the reviewer for the time taken to review our work, and for the valuable suggestions given. In this revised manuscript, we have rechecked and proofread all the spellings/typographical and grammatical/technical errors, we took the help of "Wallace Academic Editing" for professional English editing provided by our institution. https://www.editing.tw/en/brief-introduction-wallace to polish the language and to make our manuscript better for more clarity as per the reviewer’s kind suggestions.
2)In the Results sections 3.1, results regarding the screening of a patient cohort is reported, but in Materials & Methods no description of this activity is present. In a similar way, in 3.5 section Fig. 5A shows an heatmap regarding the expression of miR’s, but no information about this analysis is provided in Materials & Methods.
A2: We appreciate the reviewer’s suggestions and concern. We have rechecked and provided all information about the patient's cohort and in-silico analysis details in the main text at materials and method section. Please kindly see our materials and method section in the main text on line 134-146.
2.3. In-silico Data Acquisition and Analysis
To assess, the biological effect of the KRAS mutation in CRC patients, the data from The Cancer Genome Atlas Program (TCGA)-CRC and Metastatic CRC (MSKCC, Cancer Cell 2018) dataset contains survival data with the clinical information, KRAS-mutation, and mRNA-expression values were downloaded and analysed from cBioPortal for Cancer Genomics (http://www.cbioportal.org/). The genome-wide miRNA expression level of the 30-CRC patient's data between KRAS-wild type and KRAS-mutant tumors samples were download from the publically available Gene Ex-pression Omnibus (GEO) database (https://www.ncbi.nlm. nih.gov/geo/), and the ac-cession numbers are GSE66548[22]. We used the edgeR and pheatmap package from the R software (R version 3.3.2) to perform the differential analysis (http://www.bioconductor.org/packages/release/bioc/html/edgeR.html) and heatmap cluster analysis (https://cran.r-project.org/web/packages/pheatmap/index.html).
3)The Discussion has to be more focused on the results of the research, avoiding the repetition of concepts already reported in the
A3: We are grateful for the reviewer's comment. As per the suggestions, we have changed and fixed the issue in the discussion section pointed by the reviewer. Please refer to the discussion in our revised manuscript in line 460-539
- Discussion
Cancers of the GI tract are a major cause of mortality in patients, mainly CRC [2]. CRC is the third ubiquitous cancer worldwide and in Asia [3]. Intrinsic and acquired therapy resistance are the main hurdles to overcome for maximizing the benefits of therapy. Notably, the KRAS mutation is associated with low overall survival in pa-tients with advanced-stage CRC who are treated with cetuximab after radiation ther-apy [31]. The use of anti-EGFR therapy including cetuximab and panitumumab had significantly improved the survival of KRAS wild-type MSKCC patients, but for the KRAS mutant group, it seems ineffective [29]. Other studies have hypothesized that the anti-EGFR effect is Ras-dependent [32]. Overexpression of the Ras-signaling pathway in KRAS-mutant CRC disrupts the downstream signal transduction of anti-EGFR therapy that render KRAS-mutant CRC resistant to cetuximab [33]. Therefore, KRAS-mutant CRC is a good research model to study the biology and mechanism of anti-EGFR therapy as well as to establish and discovery of new therapeutic strategies for reverting the resistance of KRAS-mutant CRC to cetuximab.
In this present study, we did in-silico analysis to investigate the expression of miRNA’s in KRAS-mutant vs KRAS-wild CRC samples, and observed miR-193a-3p is significantly suppressed in mutant KRAS samples as compared to control (wild) sam-ples. In-vitro analysis of gain/loss of function of miR-193a-3p in the CRC cells demon-strated its opposite correlation with the expression of KRAS. The converse miR-193a-3p/KRAS relationship is associated with enhanced oncogenicity of CRC cells. Whereas, when 4-AAQB is introduced, it modulates the re-expression of miR-193a-3p, akin to short-interfering RNA, effectively reduced the expression of KRAS, results in suppression of CRC oncogenicity and tumorigenesis. Tumor suppressive effect of miR-193a-3p in cancer is well studied, especially its anti-lung cancer effect via target-ing KRAS. Studies reported and confirmed KRAS is a potential target of miR-193a-3p. Therefore, induction of miR-193a-3p expression via 4-AAQB might modulate the an-ti-CRC effect. Previously our group reported the pivotal 4-AAQB’s anti-CRC role on CRC cell lines through impeding the formation of CRC cancer stem cells and ROS (Re-active oxygen species) oxidative stress, resultant in the modulation of CRC cells innate or acquired insensitivity towards chemotherapy.
In this study, we examined, that the association of KRAS-mutation status with overall survival among CRC patients (Fig. 1) and showed the resistance of KRAS-mutated CRC cells to cetuximab (Fig. 2A). The cell viability assay showed that KRAS-mutant SW1463 cells were cetuximab-resistant in comparison with Caco-2 cells (Fig. 2B). Furthermore, no change in the clonogenic and invasive properties of SW1463 cells was noted after cetuximab treatment, indicating their resistance to cetuximab (Fig. 2D, F).
Drug resistance caused by the overexpression of the Ras-signaling pathway fre-quently occurs in KRAS-mutant CRC [34]. Inhibiting the Ras-signaling is crucial to overcome its drug resistance [35]. Thus, we hypothesized that modulation of the Ras-signaling pathway by using 4-AAQB, could resensitize KRAS-mutant CRC cells to anti-EGFR therapy. Our present work is the first time to suggest that 4-AAQB treat-ment effectively targets KRAS-mutant and wild-type CRC cells and resensitizes them to cetuximab therapy. The 4-AAQB treatment effectively reduced the viability of CRC cells dose-dependently along with the reduction in the oncogenic property and tu-mor-sphere generation abilities of CRC cells (SW1463 and Caco-2) (Fig. 3). Protein ex-pression revealed that 4-AAQB treatment reduced the phosphorylation activation of key targets participating in the Raf/Mek/Erk pathways and RAS cascade (Fig. 3H) [36]. This is consistent with our previous study, which showed that 4-AAQB suppressed tumorigenesis and inhibited cancer stem cell-like phenotype through multiple signal-ing pathways, including JAK-STAT and Wnt//β-catenin [37]. Moreover, we demon-strated that 4-AAQB induced the apoptosis of CRC cells through the suppression of Ras-signaling pathway via modulating the expression of cl-Caspase3/9 and cl-PARP [38]. Furthermore, few studies have reported that increased autophagy is associated with anti-EGFR drug resistance [39]. Our previous study reported the effect of 4-AAQB on ovarian cancer [40]; 4-AAQB could disrupt autophagy through the inhibition of the upstream signal of ATG (autophagy-related gene).
Importantly, for the first time as of our knowledge to demonstrate that through the inhibition of the Ras-signaling pathway, 4-AAQB can inhibit cancer cell growth and proliferation and induces apoptosis. Ras-signaling is involved in anti-EGFR ther-apy because Ras-protein overexpression causes the inhibitory signal from EGFR [41]. For validation, we comparatively assessed the effect of 4-AAQB and cetuximab as sin-gle-drug and in combination in KRAS wild-type Caco-2 cells human CRC cell and KRAS-mutant SW1463 human CRC cells (Fig. 4). Our result demonstrated that the combination treatment with 4-AAQB and cetuximab could synergistically overcome the resistance of KRAS-mutant CRC cells (SW1463) and KRAS-wild (Caco-2) cells to cetuximab. These findings illustrate that 4-AAQB inhibits Ras-signaling through phosphorylation modification of the RAF/MEK/ERK and Ras cascade, enhancing the potency of anti-EGFR therapy. Furthermore, regarding epigenetic factors controlling the severity of KRAS-mutant CRC cells, we found that miR-193a-3p expression was effectively downregulated in patients with KRAS-mutant CRC (Fig. 5). Furthermore, a previous study also suggested that miR-193-3p inhibition in CRC patients regulated tumor progression [42]. Conversely, miR-193a-3p overexpression inhibited tumor pro-gression. Both in-vitro and in-vivo, 4-AAQB alone or in combination with cetuximab increased miR-193a-3p expression in the xenograft mice model (Fig. 5 and 6).
As EGFR is the upstream initiator of the Ras pathway, inhibiting EGFR cannot ef-fectively inhibit downstream signals when Ras drives the whole pathway [43]. Resen-sitization of KRAS-mutant CRC to cetuximab after turning off Ras signaling has been reported in several studies [44]. 4-AAQB which shares its structure to antroquinonol (AQ), has demonstrated good anticancer potency in CRC [45]. Both 4-AAQB and AQ suppresses Ras-signaling, but which drug has a stronger effect on CRC remains elusive. We hope that use of the natural compounds of 4-AAQB would benefit many patients with CRC harboring KRAS mutations.
4)Introduction.
Since the authors affirm that Antrodia Cinnamomea is present only in Taiwan, they have to discuss the possibility to synthetize 4-Acetyl-Antroquinonol B or to grow the fungus in other countries. Otherwise, the impact of their results is significantly reduced.
A4: We thank the Reviewer for bringing up this good point. Yes, our collaborators who are expert in synthesizing small molecules are trying to synthesize 4-AAQB so that the dependency of this product on the availability of Antrodia cinnamomea can be reduced.
5)INTRODUCTION
The sentence “One of the classic examples of KRAS mutation causing drug resistance in colorectal cancer is Cetuximab, an anti-EGFR targeting monoclonal antibody” is wrong because Cetuximab is not a KRAS mutation.
A5: We are grateful for the reviewer’s insight. We are sorry for the confusion; we have rephrased the sentence and amended the required changes. Please kindly see our revised introduction section in the main text on line 55-102.
6)MATERIALS & METHODS.
- The section 2.2 can be included in 2.1, since it refers to specific products.
Please kindly see our revised MATERIALS & METHODS section
- Section 2.5. More information about cell detachment has to be added.
Please kindly see our revised MATERIALS & METHODS section
- Since Section 2.10 refers to an “in vivo” analysis, it must to be placed after 2.11.
A6: We appreciate the reviewer’s suggestions. We have rechecked and corrected the sections according to the comment that has been made, thanks. Please kindly see our newly Figure 6.
- The effect on apoptosis has been evaluated by determining caspase 3 and 9 in Western Blot. The authors have to specify if they analysed pro-caspases or cleaved caspases that are the active form of these enzymes.
A6: We thank the Reviewer for bringing up this good point, and we agree with this. We checked the expression of cleaved cl-Caspase3 and cl-PARP1 through western blot as suggested by the reviewer, we have now revised the result and figures. Please kindly see our newly Figure 4 and Figure 6, and results in the main text.
7)FIGURES AND RESULTS.
Figure 3b. The effect of 4-AAQB seems to be dose-dependent, but no information is reported with regard to the time-dependency.
A7: We are grateful for the reviewer's comment. We have rechecked and provided the time-dependent growth inhibition curve. Please kindly see our newly Figure 3.
8)Figure 4f. Why in the combination treatment 9 µM 4-AAQB has been used instead of 12 µM?
A8: We appreciate the reviewer’s suggestions. For combination treatment we used a lower dose of 4-AAQB to evaluates its potential in sensitizing the Cetuximab in low dose and modulating its Anti-CRC effect. Please kindly see our newly Figure 4.
9)Figure 5b. Statistical analysis is not reported even if in the text it is affirmed that the expression of 25 miR’s is significantly lower in KRAS mutant CRC patients.
A9: We are grateful for the reviewer’s comment. We are sorry for our oversight; we have amended the required changes and showed the statistical analysis in Figure. Please kindly see newly Figure 5.

Round 2
Reviewer 1 Report
The authors answered in all of my requests.
Author Response
Response to Reviewers:
Reviewer #1:
Q: The authors answered in all of my requests.
A: We thank the reviewers and feel overwhelmed that we answered all the questions raised.

Reviewer 2 Report
In this research paper, the authors investigated the effect of 4-AAQB in colorectal cancer cell lines that harbor KRAS mutations and cells that express wild type KRAS protein. Using a variety of assays, they showed that KRAS mutant SW1463 cells were resistant to cetuximab therapy; however, treatment with 4-AAQB resensitized CRC cells towards cetuximab therapy. This was associated with reduced KRAS signaling pathway in CRC cells, and reduced colony forming ability. Such an effect was not observed in cells containing wild KRAS. Further analysis revealed that miR-193a-3p was significantly down-regulated in the KRAS-mutated patient samples compared to KRAS-wild CRC patients. Overexpression of mIR-193a-3p resulted in a significant reduction of oncogenecity. They suggest that KRAs is a target for miR-193-3p. The in vivo treatment with 4-AAQB in combination with Cetuximab reduced the expression of EGFR, pMEK, pERK, c-RAF/p-cRAF signaling as well as induced the expression of plasma miR-193a-3p level. The authors conclude that 4-AAQB inhibit Ras signaling in CRC cell and re-sensitizes treatment with Cetuximab.
Author Response
Response to Reviewers:
Reviewer #2 (Comments to the Author):
Q: We would like to thank all the reviewers for the thorough reading of our manuscript as well as their valuable comments. We have followed the reviewer's comments thoroughly and feel that they have further helped in strengthening the manuscript.
Comments and Suggestions for Authors
In this research paper, the authors investigated the effect of 4-AAQB in colorectal cancer cell lines that harbor KRAS mutations and cells that express wild type KRAS protein. Using a variety of assays, they showed that KRAS mutant SW1463 cells were resistant to cetuximab therapy; however, treatment with 4-AAQB re-sensitized CRC cells towards cetuximab therapy. This was associated with reduced KRAS signaling pathway in CRC cells, and reduced colony forming ability. Such an effect was not observed in cells containing wild KRAS. Further analysis revealed that miR-193a-3p was significantly down-regulated in the KRAS-mutated patient samples compared to KRAS-wild CRC patients. Overexpression of mIR-193a-3p resulted in a significant reduction of oncogenecity. They suggest that KRAs is a target for miR-193-3p. The in vivo treatment with 4-AAQB in combination with Cetuximab reduced the expression of EGFR, pMEK, pERK, c-RAF/p-cRAF signaling as well as induced the expression of plasma miR-193a-3p level. The authors conclude that 4-AAQB inhibit Ras signaling in CRC cell and re-sensitizes treatment with Cetuximab.
A: We are grateful to the Reviewer and feel overwhelmed that we answered all the questions raised.

Reviewer 3 Report
The authors adressed some of the critical points present in the manuscritp improving it, but several mistakes in English style and grammar are still present also in the revised parts of the mannuscript. For this reason a further careful revision is necessary.
Moreover, some concepts are repeated several times (in the Introduction and in the Discussion), even in the same paragraph.
The authors indifferently use the concept of vitality and proliferation, but SRB test evaluates the amount of proteins and this values can reflect the number of cells, not their viability, that needs to be evaluated by specific analyses, as the release of lactate dehydrogenase in the culture medium.
Changes in phosphorilation level of proteins cannot be considered changes in the expression, since it is a post-translational modification affecting protein activity.
Author Response
Response to Reviewers:
Reviewer #3 (Comments to the Author):
Q1: Reviewer #3: The authors’ addressed some of the critical points present in the manuscript improving it, but several mistakes in English style and grammar are still present also in the revised parts of the manuscript. For this reason, a further careful revision is necessary.
A1: We sincerely thank the reviewer for the time taken to review our work, and for the valuable suggestions given. In this revised manuscript, we have proofread all the spellings/typographical and grammatical/technical errors; we took the help of "Wallace Academic Editing" for professional English editing provided by our institution. https://www.editing.tw/en/brief-introduction-wallace to polish the language and to make our manuscript better for more clarity as per the reviewer’s kind suggestions.
Q2: Reviewer #3: Moreover, some concepts are repeated several times (in the Introduction and in the Discussion), even in the same paragraph.
A2: We sincerely appreciate reviewers’ insightful comments and suggestions for revising the manuscript. We have rechecked and amended the issue related to the repetition of the concept in the introduction and discussion. Please kindly see our updated discussion section in the main text on line 460-523.
Q3: Reviewer #3: The authors indifferently use the concept of vitality and proliferation, but SRB test evaluates the amount of proteins and this values can reflect the number of cells, not their viability, that needs to be evaluated by specific analyses, as the release of lactate dehydrogenase in the culture medium.
A3: We are grateful to the Reviewer for inviting us to clarify this important point about cell viability and proliferation by SRB (Sulforhodamine B) assay, and for the valuable suggestions given. In this current study, we detected the effect of 4-AAQB on the CRC cell toxicity by using SRB assay. Is it being used extensively for large scale screening or discovery of novel anticancer drugs by the National Cancer Institute (NCI) since its discovery[1], As SRB assay is a rapid and sensitive colourimetric method for measuring the drug-induced cytotoxicity in both attached and suspension cell cultures. Many studies have reported it can be successfully used to measure both proliferation and viability of cancer cells. As per the study conducted by Kasinski et al., 2015[2], they applied SRB to evaluate the effect of gene expression modulation (by knockdown or overexpression of gene), as well as to study the effect of miRNA replacement on the cell proliferation by using SRB assay on non-small cell lung cancer. Besides its multi-use SRB assay has many advantages compared to LDH assay, mainly the SRB assay is simple, fast, and sensitive. It provided good linearity with cell number, permitted the use of saturating dye concentrations, is less sensitive to environmental fluctuations, is independent of intermediary metabolism, and provided a fixed endpoint that does not require a time-sensitive measurement of initial reaction velocity. The reproducibility of SRB assay is very high. Whereas, the LDH assay’s major limitation is that serum and some other compounds have inherent LDH activity[1]. For example, the fetal calf serum has extremely high background readings. Therefore, this assay is limited to serum-free or low-serum conditions, limiting the assay culture period (depending on your cells’ tolerance to low serum) and reducing the scope of the assay as it can no longer allow determination of cell death caused under normal growth conditions (i.e. in 10% fetal calf serum). At a minimum, we should always first test the assay with an unused aliquot of the media you intend to use and compare the reading to that from media lacking supplements (e.g. straight DMEM) which is tricky and time cumbersome [3].
Reference:
- Skehan, P.; Storeng, R.; Scudiero, D.; Monks, A.; McMahon, J.; Vistica, D.; Warren, J.T.; Bokesch, H.; Kenney, S.; Boyd, M.R.J.J.J.o.t.N.C.I. New colorimetric cytotoxicity assay for anticancer-drug screening. 1990, 82, 1107-1112.
- Kasinski, A.L.; Kelnar, K.; Stahlhut, C.; Orellana, E.; Zhao, J.; Shimer, E.; Dysart, S.; Chen, X.; Bader, A.G.; Slack, F.J.J.O. A combinatorial microrna therapeutics approach to suppressing non-small cell lung cancer. 2015, 34, 3547-3555.
- Aslantürk, Ö.S. In vitro cytotoxicity and cell viability assays: Principles, advantages, and disadvantages. InTech: 2018; Vol. 2.
Q4: Reviewer #3: Changes in phosphorylation level of proteins cannot be considered changes in the expression, since it is a post-translational modification affecting protein activity.
A4: We thank the Reviewer for bringing up this good point and inviting us to clarify this important point about phosphorylation. Protein phosphorylation is one of the most common and important post-translational modifications (PTMs) involved in the regulation of multiple biological processes and cellular processes, including cell cycle, growth, apoptosis and signal transduction pathways. [4]. As many enzymes and receptors are activated/deactivated by phosphorylation and dephosphorylating events through kinases and phosphatases [4,5]. Mutations or defects in regulatory mechanisms can lead to aberrant activation or dysregulation of kinase signaling pathways and this is the basis of oncogenesis for multiple tumors [6-9]. In our current study, we applied western blot analysis to evaluate the effect of 4-AAQB on the protein activity, and 4-AAQB effect on inhibiting the expression of gene expression (active phosphorylated state) and thus resultant in the regulation of downstream signalling, which in turn if remains un-checked can lead to CRC progression.
Reference:
- Li, X.; Wilmanns, M.; Thornton, J.; Köhn, M. Elucidating human phosphatase-substrate networks. Science signaling 2013, 6, rs10.
- Sacco, F.; Perfetto, L.; Castagnoli, L.; Cesareni, G.J.F.l. The human phosphatase interactome: An intricate family portrait. 2012, 586, 2732-2739.
- Harsha, H.C.; Pandey, A. Phosphoproteomics in cancer. Molecular oncology 2010, 4, 482-495.
- Hanahan, D.; Weinberg, R.A.J.c. Hallmarks of cancer: The next generation. 2011, 144, 646-674.
- Hynes, N.E.; MacDonald, G.J.C.o.i.c.b. Erbb receptors and signaling pathways in cancer. 2009, 21, 177-184.
- Sharma, A.; Tan, T.H.; Cheetham, G.; Scott, H.S.; Brown, M.P.J.J.o.T.O. Rare and novel epidermal growth factor receptor mutations in non—small-cell lung cancer and lack of clinical response to gefitinib in two cases. 2012, 7, 941-942.

Round 3
Reviewer 3 Report
The Authors partially replied to the Referee’s comments.
With regard to the answer to Q3, it is well known that SRB assay, forming an electrostatic complex with the basic amino acid residues of proteins, provides information about cellular protein content and, in consequence, about cell number. Anyway, in case of observed decreased values, it is impossible to distinguish between cytotoxic or cytostatic effect. This information can be obtained, as previously reported, by specific necrosis test. The accuracy of LDH results can be obtained by subtracting to the sample values the background due to the presence of serum in the culture medium.
For this reason, I suggest the Authors to refer to cell proliferation, not to cell viability.
With regard to the answer to Q4, it is well known the role played by phosphorylation in modulating the activity of several proteins, but the statement “Western blot analysis of tumour samples collected from all groups demonstrated the changes in the expression of oncogenic markers (p-EGFR), p-ERK, p-MEK, c-RAF/BRAF and p-c-RAF/BRAF” has to be changed because the phosphorylation level of the proteins is independent from their expression.
I suggest the Authors to speak of “expression”, with regard to the non-phosphorylated forms, and of phosphorylated protein levels.
In the manuscript some mistakes are still present (for example: line 136 To assess, the biological effect; line 191 aggregation for a week, Cells (diameter >50 μm); line 473 we examined, that the association of KRAS).
Author Response
Point-by-point responses to reviewer's comments:
We would like to thank the reviewer for the thorough reading of our manuscript as well as their valuable comments. We have followed the reviewer's comments thoroughly and feel that they have further helped in strengthening the manuscript.
Reviewer #3 (Comments to the Author):
Q1: Reviewer #3: With regard to the answer to Q3, it is well known that SRB assay, forming an electrostatic complex with the basic amino acid residues of proteins, provides information about cellular protein content and, in consequence, about cell number. Anyway, in case of observed decreased values, it is impossible to distinguish between cytotoxic or cytostatic effect. This information can be obtained, as previously reported, by specific necrosis test. The accuracy of LDH results can be obtained by subtracting to the sample values the background due to the presence of serum in the culture medium.
For this reason, I suggest the Authors to refer to cell proliferation, not to cell viability.
A1: We sincerely again thank the reviewer for the time taken to review our work, and for the valuable suggestions given, the suggestions will further strengthen our manuscript. We have rechecked and referred to cell proliferation throughout the manuscript. Please kindly see our revised manuscript text, on lines 276-303.
Q2: Reviewer #3: With regard to the answer to Q4, it is well known the role played by phosphorylation in modulating the activity of several proteins, but the statement “Western blot analysis of tumour samples collected from all groups demonstrated the changes in the expression of oncogenic markers (p-EGFR), p-ERK, p-MEK, c-RAF/BRAF and p-c-RAF/BRAF” has to be changed because the phosphorylation level of the proteins is independent from their expression.
I suggest the Authors to speak of “expression”, with regard to the non-phosphorylated forms, and of phosphorylated protein levels.
A1: We appreciate the reviewer’s suggestions. We agree with the suggestions given, we have rechecked and change the sentence “in place of expression we used non-phosphorylated forms, and of phosphorylated protein levels.” Please kindly see our revised manuscript text, on lines 437-442.
Q3: Reviewer #3: In the manuscript some mistakes are still present (for example: line 136 To assess, the biological effect; line 191 aggregation for a week, Cells (diameter >50 μm); line 473 we examined, that the association of KRAS).
A3: We thank the reviewer for the insightful comment. We have rechecked and reworded the text as per the reviewer's suggestion in our revised manuscript. Please kindly see our revised manuscript text, on lines 136-147, line 190-193 and line 473-474.
